# Monitoring the autophagy-endolysosomal system using monomeric Keima-fused MAP1LC3B

**Hideki Hayashi**[1], **Ting Wang**[1], **Masayuki Tanaka**[1], **Sanae Ogiwara**[1], **Chisa Okada**[1], **Masatoshi Ito**[1], **Nahoko Fukunishi**[1], **Yumi Iida**[1], **Ayaka Nakamura**[1], **Ayumi Sasaki**[1], **Shunji Amano**[1], **Kazuhiro Yoshida**[1], **Asako Otomo**[2,3,4], **Masato Ohtsuka**[5], **Shinji Hadano**[2,3,4,6]*

1 Support Center for Medical Research and Education, Isehara Research Promotion Division, Tokai University, Isehara, Kanagawa, Japan, 2 Molecular Neuropathobiology Laboratory, Department of Molecular Life Sciences, Tokai University School of Medicine, Isehara, Kanagawa, Japan, 3 The Institute of Medical Sciences, Tokai University, Isehara, Kanagawa, Japan, 4 Micro/Nano Technology Center, Tokai University, Hiratsuka, Kanagawa, Japan, 5 Genetic Engineering and Genome Editing Laboratory, Department of Molecular Life Sciences, Tokai University School of Medicine, Isehara, Kanagawa, Japan, 6 Research Center for Brain and Nervous Diseases, Tokai University Graduate School of Medicine, Isehara, Kanagawa, Japan

* shinji@is.icc.u-tokai.ac.jp

**Data Availability Statement:** All relevant data are within the manuscript and its Supporting Information files.

## Abstract

The autophagy-endolysosomal pathway is an evolutionarily conserved degradation system that is tightly linked to a wide variety of physiological processes. Dysfunction of this system is associated with many pathological conditions such as cancer, inflammation and neurodegenerative diseases. Therefore, monitoring the cellular autophagy-endolysosomal activity is crucial for studies on the pathogenesis as well as therapeutics of such disorders. To this end, we here sought to create a novel means exploiting Keima, an acid-stable fluorescent protein possessing pH-dependent fluorescence excitation spectra, for precisely monitoring the autophagy-endolysosomal system. First, we generated three lines of transgenic (tg) mouse expressing monomeric Keima-fused MAP1LC3B (mKeima-LC3B). Then, these tg mice were subjected to starvation by food-restriction, and also challenged to neurodegeneration by genetically crossing with a mouse model of amyotrophic lateral sclerosis; i.e., SOD1$^{H46R}$ transgenic mouse. Unexpectedly, despite that a lipidated-form of endogenous LC3 (LC3-II) was significantly increased, those of mKeima-LC3B (mKeima-LC3B-II) were not changed under both stressed conditions. It was also noted that mKeima-LC3B-positive aggregates were progressively accumulated in the spinal cord of SOD1$^{H46R}$;mKeima-LC3B double-tg mice, suggestive of acid-resistance and aggregate-prone natures of long-term overexpressed mKeima-LC3B *in vivo*. Next, we characterized mouse embryonic fibroblasts (MEFs) derived from mKeima-LC3B-tg mice. In contrast with *in vivo*, levels of mKeima-LC3B-I were decreased under starved conditions. Furthermore, when starved MEFs were treated with chloroquine (CQ), the abundance of mKeima-LC3B-II was significantly increased. Remarkably, when cultured medium was repeatedly changed between DMEM (nutrient-rich) and EBSS (starvation), acidic/neutral signal ratios of mKeima-LC3B-positive compartments were rapidly and reversibly shifted, which were suppressed by the CQ

**Funding:** This study was supported by a Grant-in-Aid for Challenging Exploratory Research (24650189 to S.H.) and partly by a Grant-in-Aid for Scientific Research (B) (23300129, 26290018 to S. H.) from the Japanese Society for Promotion of Science (JSPS).

**Competing interests:** The authors have declared that no competing interests exist.

treatment, indicating that intraluminal pH of mKeima-LC3B-positive vesicles was changeable upon nutritional conditions of culture media. Taken together, although mKeima-LC3B-tg mice may not be an appropriate tool to monitor the autophagy-endolysosomal system *in vivo*, mKeima-LC3B must be one of the most sensitive reporter molecules for monitoring this system under *in vitro* cultured conditions.

## Introduction

The autophagy-endolysosomal pathway is an evolutionarily conserved degradation system that is tightly linked to a wide variety of physiological processes [1, 2]. Three different forms of the autophagic pathways; i.e., macroautophagy, microautophagy, and chaperon-mediated autophagy are currently documented. In particular, macroautophagy (hereafter referred to as "autophagy") together with the endolysosomal pathway plays a crucial role in the removal and degradation not only of cytoplasmic long-lived as well as misfolded proteins but also of damaged or superfluous organelles through a sequential step comprising the autophagosome formation, maturation (fused with endosomes and/or lysosomes) and degradation within autolysosomes/lysosomes [1].

A plethora of studies has been demonstrated that dysfunction of the autophagy-endolysosomal system is associated with many pathological conditions such as cancer, inflammation and neurodegenerative diseases [2–6]. In order to uncover the contribution of this system to the pathogenesis, and then to develop the therapeutic treatment for such disorders, proper monitoring of the autophagy-endolysosomal activities is extremely important [7]. Thus far, microtubule-associated protein 1 light chain 3 (LC3), whose lipidated form (LC3-II) is highly enriched onto autophagosomal membrane during entire stages of autophagy; from the formation to maturation of autophagosomes, has reliably been used for such purposes [7, 8]. In particular, LC3 tagged with a green fluorescent protein, GFP-LC3 [9], has widely been utilized in not only *in vitro* culture experiments but also *in vivo* animal studies. Indeed, we and others have previously reported that GFP-LC3 can be used to monitor the autophagic status as well as disease progression in a GFP-LC3-expressing animal model of amyotrophic lateral sclerosis (ALS); i.e., human mutant SOD1-expressing transgenic mice [10, 11].

Despite of such usefulness of GFP-LC3, there are some shortcomings to monitor the flux throughout the entire autophagic system due to the reason that the fluorescence of GFP diminishes under lysosomal acidic conditions. To overcome such weaknesses, several groups have generated a series of acid-resistant and -sensitive tandem fluorescent proteins fused with LC3, such as RFP-GFP-LC3 [12–14], mCherry-EGFP-LC3 [15] and GFP-LC3-RFP-LC3ΔG [16], as an alternate to GFP-LC3. However, since quenching the fluorescence of GFP under the course of gradual maturation and/or acidification of intralumenal endosomes and/or autophagosomes slowly and incompletely occurs, the sensitivity for monitoring the maturation of endosomes and autophagosomes using these tandem fusion LC3 proteins might still be limited. Another type of the pH-sensitive and acid-resistant tandem probes such as mTagRFP-mWasabi-LC3 [17] and pHluorin-mKate2-tagged LC3 [18], both of which seemed to be more appropriate to monitor the maturation of autophagosomes, was also reported, but their sensitivities for monitoring the acidification of particular vesicles have yet to be thoroughly investigated.

Recently, a fluorescent variant of a protein from the stony coral *Montipora*, called Keima, has been developed [19], and applied to monitor the autophagy activity [20]. With the use of

monomeric Keima-Red (mKeima-Red), which has variable excitation spectra in a pH-dependent fashion and shows proteolytic resistance in acidic compartments, the entire autophagy flux can be more efficiently monitored by a single fluorescence protein alone [20]. When mKeima fuses with a tandem repeat sequence of cytochrome c oxidase subunit 8 (mt-mKeima), it can also be specifically recruited to mitochondria, allowing a sensitive and selective monitoring of mitophagy with the simultaneous use of other green emitting fluorophores in cells [20]. Thus, if mKeima could be properly targeted to the membrane of autophagosomes as well as endocytic vesicles by fusing with selectively localized molecules such as LC3, a more selective and sensitive monitoring of the vesicular maturation and acidification would become possible. Furthermore, like GFP-LC3, in order to measure the autophagic-flux in tissues and/or cells by Western blotting with the use of appropriate antibodies [10], mKeima-fused LC3 could be utilized as an "autophagomometer" [8] much easier than endogenous LC3.

In this study, by exploiting merits of mKeima and LC3 combined, we newly developed a single fluorescent protein-based and pH-sensitive probe, mKeima-LC3B, as a novel analytical tool for the autophagy-endolysosomal system. We generated transgenic (tg) mice that were expressing mKeima-LC3B. To assess their usefulness, we characterized biochemical as well as histological phenotypes of mKeima-LC3B-tg mice under stressed conditions; i.e., starvation or neurodegeneration *in vivo*. Further, we investigated cellular responses to the starvation using mouse embryonic fibroblasts (MEFs) derived from these animals *in vitro*.

## Materials and methods

### Plasmid construction

We generated pCAG_mKeima_LC3B plasmid to express mKeima fused to human MAP1LC3B (LC3B) under control of the chicken β-actin (CAG) promoter in mammalian cells. The mKeima and LC3B sequences were amplified from pmKeima-S1 (MBL) and pEGFP-LC3B [21] by PCR using primers with site-specific restriction sites, respectively (mKeima: XhoI-AgeI-kozac-mKeima_F; 5'-gatctcgagaccggtccaccatggtgagtgtg atcgcta-3' and PacI-mKeima_R; 5'-gatcttaattaaaccgagcaaagagtggcgtg-3', LC3B: PacI-LC3B_F; 5'-gatcttaattaatatgccgtcggagaagacct-3' and SmaI-BsiWI-LC3B_R; 5'-gatcccgggcgtacgttacactgacaatttcatcccga-3'). These amplicons were digested with proper restriction enzymes, ligated and cloned into the unique *Age*I-*Bsr*GI restriction sites of pAMF plasmid that contained [CAG_AgeI_EGFP_BsrGI_polyA] cassette [22]. Plasmid DNA for pCAG_mKeima-Red_LC3B extracted from transformed bacteria using NucleoBond Xtra Midi kit (TaKaRa) was further digested with *Not*I. Resulting *Not*I-restricted insert DNA (CAG_mKeima_LC3B_polyA) (S1 Fig) was purified by NucleoSpin Gel and PCR Clean-Up (TaKaRa) and subjected to microinjection.

### Animals

C57BL/6J (B6), BDF1, ICR and MCH(ICR) mice were purchased from CLEA Japan. We used two previously-established tg mice carrying the H46R mutation in the human *SOD1* gene [SOD1^H46R; a mouse model of amyotrophic lateral sclerosis (ALS)] [10, 21, 23] and those carrying the GFP-fused human-*MAP1LC3B* cDNA (GFP-LC3) [9]. SOD1^H46R-tg and GFP-LC3-tg mice were backcrossed to B6 mice for more than 10 generations and maintained as B6 congenic lines. Mice were housed at an ambient temperature of 23 ± 2˚C and humidity of 55 ± 15% with a 12 h light-dark cycle. Food and water were fed *ad libitum* unless otherwise noted. All animal experimental procedures were carried out in accord with the Fundamental Guidelines for Proper Conduct of Animal Experiment and Related Activities in Academic Research Institutions under the jurisdiction of the Ministry of Education, Culture, Sports,

Science and Technology (MEXT), Japan, and reviewed and approved by The Institutional Animal Care and Use Committee at Tokai University (Permitted #; 181033).

## Generation of mKeima-LC3B-expressing transgenic mice

To induce superovulation, we intraperitoneally injected each donor female B6 mouse (8–9 weeks of age) with 5 IU pregnant mare serum gonadotropin (PMSG) (Sankyo Yell), and female BDF1 mouse (8–12 weeks of age) with 7.5 IU PMSG. After 48 h, 5 IU human chorionic gonadotropin (hCG) (Sankyo Yell) was further injected intraperitoneally into each donor mouse. Oocytes that were recovered from superovulated female B6 and BDF1 mice were subjected to *in vitro* fertilization (IVF) with sperms obtained from B6 and BDF1 male mice (10 weeks of age), respectively. To create tg animals, we microinjected *Not*I-restricted insert DNA (CAG_mKeima_LC3B_polyA) into 428 B6-derived and 813 BDF1-derived fertilized eggs. Among them, 369 B6-derived and 511 BDF1-derived eggs were transplanted into the oviduct of pseudopregnant recipient ICR and MCH(ICR) mice, respectively.

To confirm the presence of transgene in the genome, we designed two sets of primer pair for PCR; Keima_LC3 fused gene (338 bp): Keima_F1; 5'-tctttgcacgagatggaatg-3' and LC3B_R1; 5'-tatcaccgggatttttggttg-3', and CAG promoter (239 bp): CAG_F2; 5'-ccgctcgacattgattattga-3' and CAG_R2; 5'-tgccaagtgggcagt ttac-3' (S1 Fig). Genomic DNA was extracted from ear tissue of mice. Standard reaction mixture for ExTaq (TaKaRa) was used for amplification. Thermal conditions used were as follows; 95˚C 5 min denaturation followed by 30 cycles of [98˚C/10 s, 56˚C/30 s, 68˚C/3 min]. PCR products were analyzed by agarose-gel electrophoresis. B6 and BDF1-derived mKeima-LC3B-tg mice were maintained as B6 background and were backcrossed to B6 mice for 4–5 generations, respectively.

## Copy number estimation

Copy numbers of the transgene in tg mice were determined by quantitative PCR. Genomic DNA of each tg mouse was obtained from tail tissues by the phenol-chloroform extraction. Quantitative PCR was performed by FastSYBR Green MasterMix by 7500 Fast Real-time PCR system (LifeTec) using following primers; for the mKeima gene: Keima_F; 5'-catctgttga gcagtgaaatag-3' and Keima_R; 5'-cgctgcttgaaggtcttctc-3', for the PGK gene as a control: PGK_F; 5'-caggactaaagatgcgtggat-3' and PGK_R; 5'-acctgc aagcgctacactt-3'. Real-time PCR was performed with thermal conditions as follows; 95˚C/2 min denaturation followed by 40 cycles of [95˚C/3 s, 60˚C/30 s]. Copy numbers of transgene were calculated by double delta Ct analysis compared with mKeima and PGK.

## Starvation *in vivo*

To study the effects of starvation *in vivo*, mice were deprived of food for 24 or 48 h. These mice were allowed to freely access drinking water.

## Generation of mKeima-LC3B-expressing ALS mouse model

We generated mKeima-LC3B-expressing SOD1[H46R] (mKeima-LC3B;SOD1[H46R]) mice by crossing male SOD1[H46R]-tg mice with female mKeima-LC3B-tg mice. The offsprings were genotyped by PCR using genomic DNA extracted from ear tissues. Primers for the SOD1 transgene were as previously described [24].

## Antibodies

Antibodies used for western blot analysis were as follows; primary antibodies included guinea-pig anti-human SQSTM1 (1:3000, PROGEN), rabbit anti-human LC3 (1:5000, MBL), rabbit anti-GAPDH (1:5000, Sigma-Aldrich) and mouse monoclonal anti-β-actin (1:10000, Sigma-Aldrich) antibodies. Secondary antibodies included horseradish peroxidase (HRP)-conjugated anti-rabbit IgG (1:5000, GE Healthcare bioscience), HRP-conjugated anti-mouse IgG (1:5000, GE Healthcare bioscience) and HRP-conjugated anti-guinea-pig IgG (1:3000, SantaCruz) antibodies.

Antibodies used for immunohistochemistry were as follows; primary antibodies included rabbit anti-human LC3 (1:1000, MBL), mouse monoclonal anti-mKeima (1:1000, DAKO), iso-type control mouse IgG2a (1:100, DAKO) and isotype control rabbit IgG (1:100, DAKO) anti-bodies. Secondary antibodies included N-Histofine® Simple Stain Mouse MAX PO (Nichirei) and EnVision+ System-HRP Labelled Polymer anti-mouse antibody (DAKO).

## Western blot analysis

Western blot analysis was performed as previously described [21] with minor modifications. Briefly, mouse tissues were homogenized in Lysis buffer [50 mM Tris-HCl (pH7.5), 150 mM NaCl, 0.1% (w/v) SDS, 0.5% (w/v) deoxycholic acid, 1% (w/v) Triton X-100, Complete Protease Inhibitor Cocktail (Roche)] using Shake Master NEO (Bio Medical Science Inc.). MEFs cultured in 3.5 cm culture dish were washed with phosphate buffered saline (PBS), and then lysed with Lysis buffer. After centrifugation at $12,000 \times g$ for 5 min, supernatant was collected. Protein concentration of these samples was determined by DC protein assay kit (Bio-Rad Laboratories). Equal amount of protein was separated on SDS-polyacrylamide gels, and then transferred onto polyvinylidene difluoride membrane (Millipore). Membranes were blocked with 5% (w/v) skimmed milk diluted in TBST [25 mM Tris-HCl (pH7.5), 150 mM NaCl, 0.1% (w/v) Tween-20] for 1 h at 37°C, and subsequently incubated with primary antibody diluted in blocking solution overnight at 4°C. After washing with TBST, membranes were incubated with HRP-conjugated secondary antibody. Immunoreactivities of primary antibodies were visualized with Immobilon-Western Chemiluminescent HRP Substrate (Millipore) and analyzed using Ez-Capture Analyzer (ATTO). Signal intensities were quantified using CS analyzer ver 3.0 (ATTO).

## Histochemistry and immunohistochemistry

Mice were anesthetized with 4% isoflurane by inhalation, and transcardially perfused with 4% paraformaldehyde/phosphate buffer (PFA/PB) (Muto Chemical). Tissues were removed and post-fixed for 48 h in 4% PFA/PB, followed by paraffin embedding. Paraffin sections were sliced on microtome at a thickness of 4–5 μm. For immunohistochemistry, tissue sections were deparaffinized and hydrated. Sections were incubated in Target Retrieval Solution (pH6.0) (DAKO) and heated at 120°C for 5 min. After cooling at room temperature (RT), sections were treated with 3% $H_2O_2$ solution to inactivate endogenous peroxidase activities. For anti-mKeima antibody, sections were blocked by the treatment with PBS (pH 7.2) containing 5% normal goat serum (NGS) for 30 min at RT, followed by incubation with primary antibody in PBS for 1 h at RT. Sections were washed with PBS for 15 min and incubated with secondary antibody for 1 h. After washing with PBS for 15 min, sections were stained with 3,3'-diamino-benzidine (DAB). For anti-LC3 antibody, sections were blocked by the treatment with TBST containing 0.05% Tween 20/5% NGS for 30 min at RT, followed by incubation with primary antibody in Antibody Diluent (DAKO) for 1 h at RT. Sections were washed with TBST for 15 min and incubated with secondary antibody for 1 h. After washing with TBST for 15 min,

sections were stained with DAB. To determine specificity of immunostaining, serial sections were similarly processed except that primary antibody was omitted. Sections were counter-stained with hematoxylin. Images were observed and captured by fluorescence microscope BX63 equipped with DP73 camera (Olympus).

## Preparation of mouse embryonic fibroblasts and cell culture

Primary MEFs were established form embryos that were obtained from wild-type, mKeima-LC3B-tg and GFP-LC3-tg mice at 13.5 days post coitus. Embryos were soaked into sterile PBS and dissected, and their visible organs were removed. Remaining skin tissues were further digested and sheared by pipetting in 1 ml of Trypsin-EDTA solution (0.5 g porcine trypsin and 0.2 g EDTA-4Na/L Hanks' Balanced Salt Solution with phenol red) (Sigma-Aldrich). After incubation for 15 min at 37°C, tissues were treated in Dulbecco's Modified Eagle Medium (DMEM) (Nacalai tesque) supplemented with 10% heat-inactivated fetal bovine serum (FBS) (Biowest), 100 U/ml penicillin and 100 μg/ml streptomycin (Sigma-Aldrich). Dissociated cells were seeded onto a 60 mm cultural dish at an appropriate cell density and cultured in the same culture medium in 5% $CO_2$ atmosphere at 37°C. Adherent cells as MEFs were used.

## Nutrient starvation *in vitro*

For starvation, MEFs established from wild-type, mKeima-LC3B-tg and GFP-LC3-tg embryos were treated with EBSS (116.4 mM NaCl, 5.4 mM KCl, 1.8 mM $CaCl_2$, 0.8 mM $MgSO_4 \cdot 7H_2O$, 1.0 mM $NaH_2PO_4 \cdot 2H_2O$, 26.2 mM $NaHCO_3$, 5.6 mM Glucose). To monitor the autophagic flux, lysosomal degradation was inhibited by addition of 50 μM chloroquine (CQ), a lysosomo-tropic agent that inhibits the lysosomal proteases, to the medium.

## Flow-cytometry

For data acquisition, we used a LSRFortessa flow cytometer (Becton Dickinson). Signals were measured using two parameters; emission filter 670/30nm (Qdot 655 filter) excited by violet laser (405 nm) and 610/20nm (PE-Tx-Red filter) by yellow-green laser (561 nm). All data were analyzed using FlowJo software (Becton Dickinson). MEFs were washed with PBS(-), treated with trypsin, and collected to a 1.5-ml tube (Sumitomo Bakelite). After centrifugation, cells were suspended with DMEM in the absence or presence of 50 μM CQ and transferred to 5mL Round Bottom Polystyrene Tube (CORNING), followed by recording data. To obtain data from MEFs under starved conditions, cultured cells in DMEM were washed with PBS(-) and trypsinized. After centrifugation, medium was replaced with EBSS, and capturing and process-ing of data were immediately (in less than 1 min) started. Four quadrants (Q1-Q4) were assigned based on the distribution of background signals observed in wild-type MEFs, in which Q4 contained the maximum number of cells within its minimum area.

## Fluorescence imaging

To examine the pH sensitivity of mKeima, MEFs expressing mKeima-LC3B were fixed with 4% PFA in PBS (-) (pH 7.5) for 15 min, followed by washing three times with PBS (-) at RT. After capturing the images for mKeima as a basal control, fixed cells were buffered at pH 4.0 (50 mM acetate buffer), pH 5.0 (50 mM acetate buffer), pH 6.0 (50 mM phosphate buffer), pH 7.0 (50 mM MOPS buffer), pH 8.0 (50 mM MOPS buffer) or pH 9.0 (50 mM bicine buffer). Then, fluorescent signals were detected. Fixed MEFs treated with EBSS (pH 7.4–7.6) were also observed. All images were captured and analyzed by ZEISS LSM880 laser scanning confocal microscope (Carl Zeiss).

## Live-cell imaging

For live-cell imaging of LC3-positive vesicular compartments in wild-type and mKeima-LC3B-tg derived MEFs, we used ZEISS LSM880 laser scanning confocal microscope equipped with a 40 x C-Apochromat water-immersion objective lens, a multi-argon laser (458, 488, and 514 nm), a DPSS laser (561 nm) and a Diode laser (405 nm) (Carl Zeiss). Images for mKeima and LysoTracker blue DND-22 (Thermo) were obtained at emission wavelengths of 616–696 nm and 406–501 nm, respectively. Images were also obtained using a 100 x Plan Apochromat (1.46 NA) oil objective on the Airyscan array detector and processed in conjunction with the Airyscan processing toolbox in the ZEN software (Carl Zeiss). Microscope incubation system was used to maintain constant environmental conditions inside the observation chamber, which was set at 37°C by injecting heated and humidified air containing 5% $CO_2$. To observe the changes in fluorescent signals for mKeima-LC3B, we repeatedly replaced culture media from DMEM to EBSS, and from EBSS to DMEM. We also measured the signals under transitional conditions from DMEM to DMEM+CQ (50 μM), DMEM to EBSS+CQ, EBSS to DMEM+CQ, EBSS to EBSS+CQ, DMEM+CQ to DMEM, DMEM+CQ to EBSS and DMEM+CQ to EBSS+CQ. Ratio (561 nm/458 nm) images of mKeima were created by ZEN software.

## High-content analysis

For live-cell time-lapse analysis, MEFs were cultured in a 96-well plate using live-cell chamber of a Cellomics Arrayscan VTI (Thermo) at 10 × magnification using Target Activation Bioapplication. During image capturing, culture plate was placed in an incubation chamber to maintain appropriate environmental conditions (37°C, 5% $CO_2$). Images for mKeima was obtained from fluorescent data by using these two excitation wavelengths (485 nm and 549 nm) and emission at > 590 nm. Ratio of signal intensity (549 nm/485 nm) was used as an index of the autophagic activity.

## Statistical analysis

Data in this study were presented as mean ± standard deviation (S.D.) or standard error of means (S.E.M.). Statistical analyses were conducted using PRISM 8 (GraphPad). Statistical significance was evaluated by ANOVA (analysis of variance) followed by Bonferroni's multiple comparison test between groups. We considered $p$-values < 0.05 to be statistically significance.

## Results

### Generation of mKeima-LC3B-expressing transgenic mice

We microinjected *Not*I-restricted insert DNA (CAG_mKeima_LC3B_polyA) into B6-derived fertilized eggs. These eggs were transplanted into the oviduct of recipient ICR mice. As a result, a total of 60 pups was obtained. Among them, 4 mice ($F_0$) were confirmed to carry the transgene in their genome. From these $F_0$ mice, 3 independent lines were successfully established (Table 1). Copy numbers of transgene in KLC3_35_105, KLC3_35_106 and KLC3_44 lines were estimated to be 80–100, 80–100 and 8 copies, respectively. To confirm the expression of protein, we next performed western blot analysis of extracts from a variety of tissues. As a control, we also used GFP-LC3-tg mice [9, 10]. Although GFP-LC3 fusion protein was ubiquitously and highly expressed in GFP-LC3-tg, mKeima-LC3B was rather preferentially expressed in the central nervous system (CNS), hearts and muscles, but not in other peripheral tissues of mKeima-LC3B-tg mice, with highest in KLC3_44 line (Fig 1A).

To obtain mouse lines that ubiquitously expressed mKeima-LC3B, we further injected the same DNA construct to BDF1-derived fertilized eggs, followed by transplantation into the

**Table 1. Summarized results of integrated copy number of transgene and the levels of protein expression in mKeima-LC3B transgenic lines.**

| Strain | Line | Copy numbers | Protein expression # | Selected line |
|---|---|---|---|---|
| B6 | KLC3_35_105 | 80~100 | n.d. | |
| | KLC3_35_106 | 80~100 | n.d. | |
| | **KLC4_44** | **~8** | **1.00** | ** |
| BDF1 | BDKLC3_7–5 | n.d. | 4.80 | |
| | BDKLC3_9–1 | n.d. | 1.40 | |
| | **BDKLC3_10–4** | **~6** | **2.09** | ** |
| | BDKLC3_11–2 | n.d. | 0.56 | |
| | BDKLC3_13–2 | n.d. | 0.66 | |
| | **BDKLC3_17–1** | **~12** | **2.58** | ** |

# Relative values to mKeima-LC3B expression in tail tissue of KLC3_44 mouse.

n.d., not determined.

**, Selected line: Transgenic mouse lines used for further analyses in this study.

oviduct of recipient MCH(ICR) mice. As a result, a total of 115 pups was obtained. Among them, 20 mice ($F_0$) were confirmed to carry the transgene. Two out of 6 lines, which expressed mKeima-LC3B in tail tissues, were further subjected to copy number analysis, revealing that BDKLC3_10–4 and BDKLC3_17–1 lines carried 6 and 12 copies of the transgene, respectively (Table 1). Next, we analyzed tissue distribution of mKeima-LC3B in these lines. Although BDKLC3_10–4 line showed a similar expression pattern as did KLC3_44, BDKLC3_17–1 line was confirmed to be ubiquitously expressed the mKeima-LC3B fusion protein (Fig 1B).

## Effects of starvation on mKeima-LC3B *in vivo*

To study the effects of starvation *in vivo*, 2 lines of mKeima-LC3B-tg mice (KLC3_44 and BDKLC3_17–1) were subjected to food-restriction study. After deprivation of food for 24 or 48 h, tissue samples were prepared from each mouse and analyzed the expression of mKeima-LC3B and endogenous LC3 using anti-LC3 antibody. By western blot analysis, no visible lipidated-form of LC3 (LC3-II) for mKeima-LC3B was observed not only in the CNS but also in heart and muscles, even though strong bands representing mKeima-LC3B-I were evident. Nonetheless, it is noted that increased level of endogenous LC3-II was observed in muscles after 48 h of starvation, indicating that autophagy is upregulated, at least, in skeletal muscles in response to food deprivation (Fig 2).

To further clarify autophagic response to starvation *in vivo*, we conducted immuno-histochemical analysis using either anti-LC3 or anti-mKeima antibody. Previously, it has been reported that food deprivation induces autophagy in the Purkinje cells and cortical neurons, [25]. However, there are no such discernable evidences in mKeima-LC3B-tg mice (S2A–S2D Fig). Instead, signals representing both endogenous LC3 and mKeima-LC3B were significantly increased in motor neurons of the spinal cord (Fig 3A and S2E Fig), indicating that motor neurons were highly responsive neurons to food deprivation *in vivo*. Regarding peripheral tissues, although there were no visible increased signals in the heart (Fig 3B and 3C), dot-like structures appeared in skeletal muscle (Fig 3D), which was consistent with the previous report [9].

## Effects of mutant SOD1 overexpression on mKeima-LC3B *in vivo*

We have previously shown that in double-tg SOD1$^{H46R}$;GFP-LC3 mice, level of a lipidated-form of GFP-LC3; i.e., GFP-LC3-II is significantly increased in the spinal cord as disease progresses [10], implying that GFP-LC3 is an excellent molecular marker to monitor the

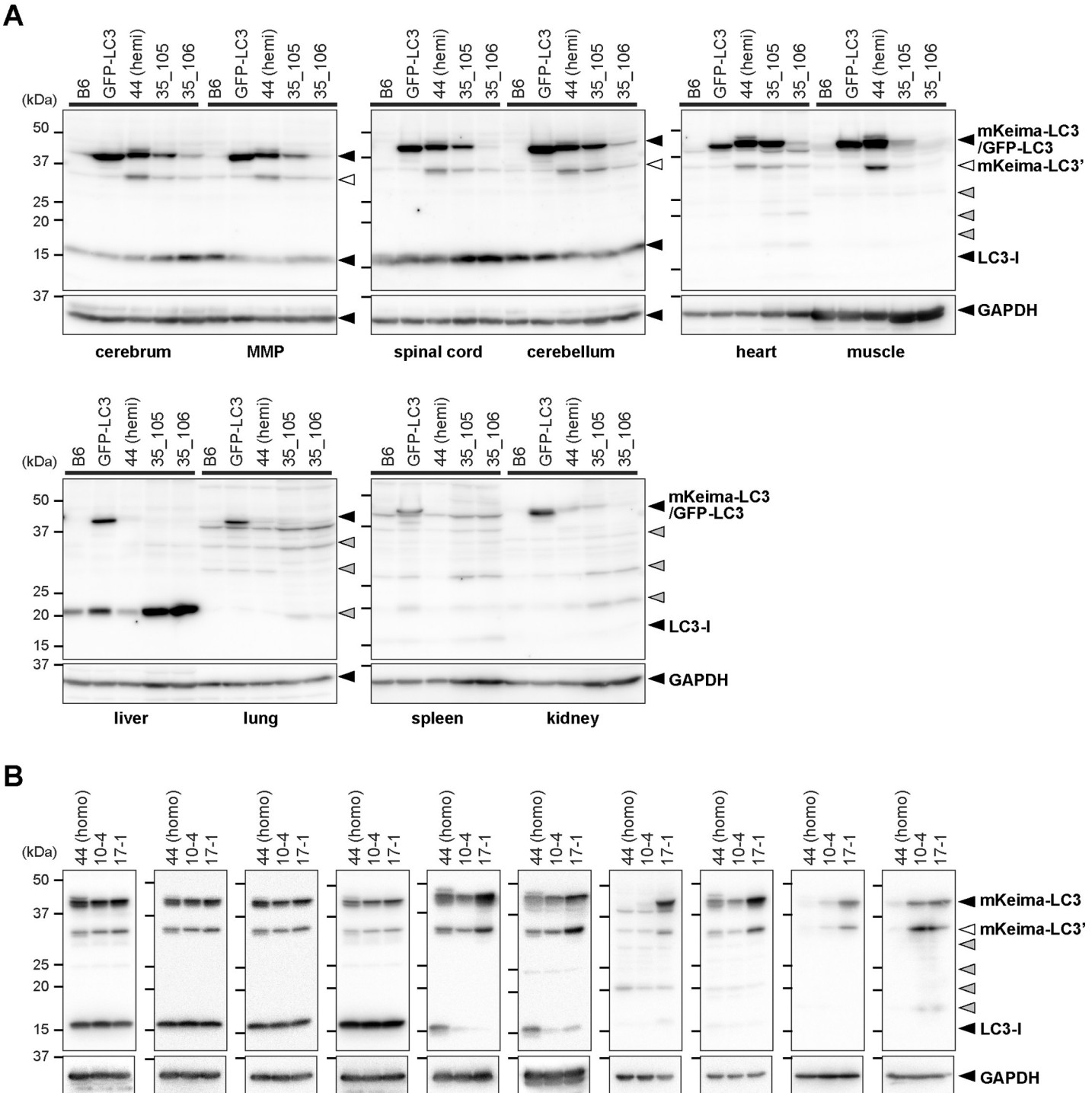

**Fig 1. Tissue distribution of mKeima-LC3B fusion protein in mKeima-LC3B transgenic mice.** (A) Western blot analysis of transgene-derived proteins (GFP-LC3 and mKeima-LC3B) and endogenous LC3 in wild-type (B6), GFP-LC3 transgenic (GFP-LC3) and 3 KLC3 lines of mKeima-LC3B transgenic [44 (hemi; hemizygote), 35_105 and 35_106] (see Table 1) mice. (B) Western blot analysis of mKeima-LC3B and endogenous LC3 in wild-type (B6), a KLC3 line of mKeima-LC3B transgenic [44 (homo; homozygote)], and 2 BDKLC3 lines of mKeima-LC3B transgenic (10–4 and 17–1) (see Table 1) mice. Tissue samples analyzed are as follows: cerebral cortex (cerebrum), midbrain + medulla + pons (MMP), spinal cord, cerebellum, heart, gastrocnemius (muscle), liver, ling, spleen, and kidney. Twenty μg of total proteins from each tissue was subjected to SDS-PAGE and analyzed by immunoblotting using anti-LC3 antibody. GAPDH was used for a loading control. Positions of mKeima-LC3B and endogenous LC3 are indicated. White and gray arrowheads shown on the right indicate a possible truncated mKeima-LC3B (indicated as mKeima-LC3') and non-specific bands, respectively. Size-markers are shown on the left.

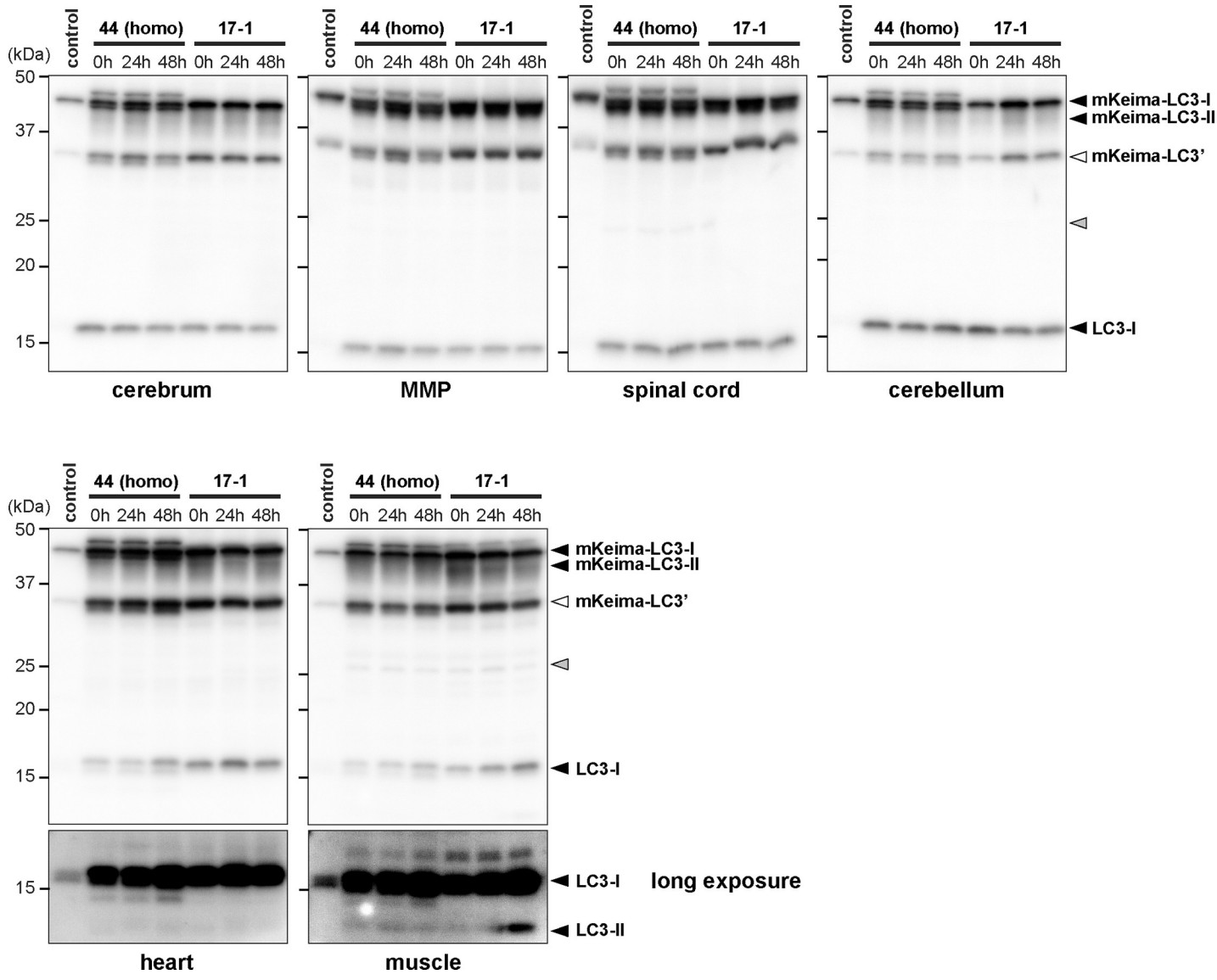

**Fig 2. Western blot analysis of LC3 in tissues from mKeima-LC3B transgenic mice under starvation conditions *in vivo*.** Two lines of mKeima-LC3B transgenic mouse (KLC3_44 and BDKLC3_17–1) were subjected to food-restriction study. Mice were fed *ad libitum* or deprived of food for either 24 or 48 h. Tissue samples analyzed are as follows: cerebral cortex (cerebrum), midbrain + medulla + pons (MMP), spinal cord, cerebellum, heart and gastrocnemius (muscle). Protein extract from whole body of a founder stillborn mouse (KLC3_56D) was also used (control). Five μg of total proteins from each tissue was subjected to SDS-PAGE and analyzed by immunoblotting using anti-LC3 antibody. Positions of mKeima-LC3B and endogenous LC3-I and LC3-II are indicated. White and gray arrowheads shown on the right indicate a possible truncated mKeima-LC3B (indicated as mKeima-LC3') and non-specific bands, respectively. Size-markers are shown on the left.

autophagy-endolysosomal system in the CNS *in vivo*. To clarify whether mKeima-LC3B would become an alternative and utilizable maker, we generated double-tg SOD1$^{H46R}$;mKeima-LC3B (KLC3_44) mice and analyzed mKeima-LC3B, endogenous LC3 and SQSTM1 in the brain and spinal cord from pre-symptomatic as well as end-stage double-tg mice. Unexpectedly, levels of a lipidated form of mKeima-LC3B (mKeima-LC3-II) were not changed by overexpression of mutant SOD1 despite of their advanced disease phenotypes (Fig 4A, 4D and 4E). By contrast, ratio of endogenous LC3-II/LC3-1, as a marker of autophagosome formation, of the same double-tg mice at end stage was significantly increased (Fig 4B, 4F and 4G). These results indicate that mKeima-LC3B does not properly convert to the lipidated form *in vivo* even

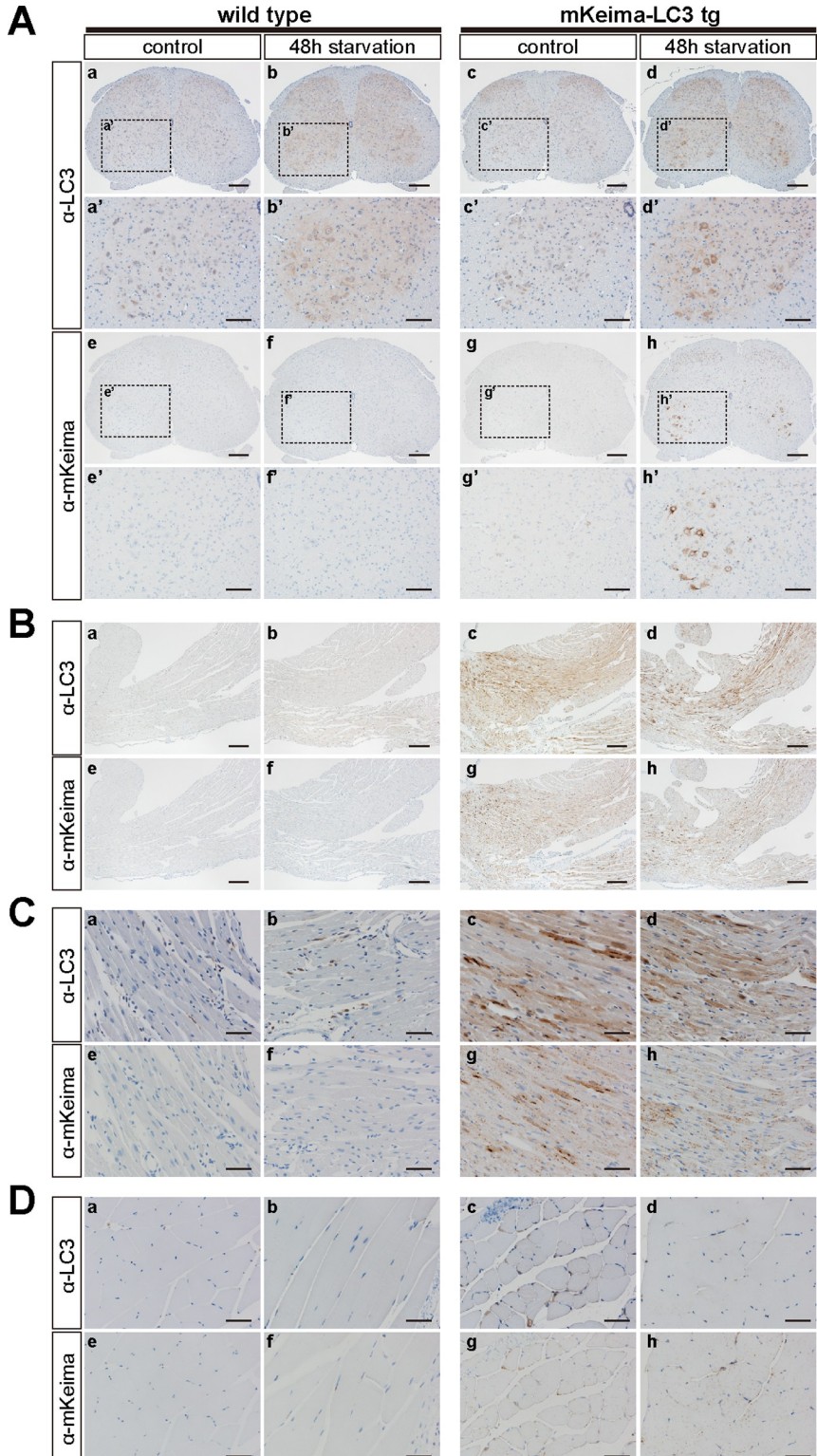

**Fig 3. Immunohistochemical analysis of LC3 in tissues from wild-type and mKeima-LC3B transgenic mice under starvation conditions** *in vivo***.** (A) Lumbar spinal cord (L4-L5), (B and C) cardiac muscles and (D) skeletal muscle (gastrocnemius), which were prepared from mKeima-LC3B transgenic (tg) mice under either fed (control) or starved (48 h) conditions, were fixed and immunostained with anti-LC3 (α-LC3) or anti-mKeima (α-mKeima) antibody. Mouse lines used in this analysis were (A) BDKLC3_17–1, (B) KLC3_44, (C) BDKLC3_17–1, and (D) BDKLC3_17–1. Scale bars indicate (A, a-h) 200 μm, (A, a'-h') 100 μm, (B, a-h) 200 μm, (C, a-h) 100 μm and (D, a-h) 100 μm.

though autophagy itself is dysregulated in the CNS, which is in stark contrast to GFP-LC3 [10]. It was also noted that SQSTM1 tended to be accumulated in the CNS of mKeima-LC3B-tg mice, suggesting that overexpression of mKeima-LC3B alone induced proteostatic stress *in vivo* (Fig 4C, 4H and 4I). Indeed, immunohistochemical analysis revealed that mKeima-LC3B-positive aggregates were progressively accumulated in the spinal cord of SOD1[H46R];mKeima-LC3B mice (Fig 5), suggestive of an aggregate-prone nature of long-term overexpressed mKeima-LC3B molecules. These data demonstrate that mKeima-fused LC3 may not be an appropriate means to monitor the autophagy-endolysosomal system under stressed conditions *in vivo*.

### Effects of starvation on mKeima-LC3B *in vitro*

Next, we investigated whether mKeima-LC3B could become a useful tool to monitor the autophagy-endolysosomal system *in vitro*. MEFs derived from wild-type, GFP-LC3-tg and mKeima-LC3B-tg (line BDKLC3_17–1) mice were subjected to autophagic-flux analysis, in which LC3-II levels in the presence or absence of lysosomal inhibitors were used as "autopha-gomometer" [8]. Under starved conditions, levels of LC3-I, including GFP-LC3-I, mKeima-LC3B-I and endogenous LC3-I, were gradually decreased over time (Fig 6A). Further, level of SQSTM1 in mKeima-LC3B-expressing MEFs was initially decreased followed by restoring back to the basal level during prolonged starvation (Fig 6A), consistent with the previous report [26]. When cells were treated with CQ, levels of GFP-LC3-II and mKeima-LC3B-II were increased (Fig 6A). Further, endogenous LC3-II in all tested cell types were comparably increased (Fig 6A). These results indicate that degree of autophagic clearance was affected by neither GFP-LC3 nor mKeima-LC3B overexpression, and thus that mKeima-LC3B-tg-derived MEFs can normally respond to starvation like do GFP-LC3-tg-derived MEFs.

To confirm whether mKeima-LC3B fusion protein would show pH-dependent fluorescence signals as anticipated [20], we analyzed the pH-sensitivity of mKeima-LC3B in mKeima-LC3B-expressing MEFs. After fixation, fluorescence signals for mKeima were first detected, and then fixed cells were treated with buffer solutions with different pH, followed by detecting fluorescence signals again. For comparison, we also obtained the images for living-cells treated with DMEM or EBSS. As a result, pH-dependent changes in mKeima-emitted fluorescence signals representing either neutral pH (excitation at 458nm) or acidic pH (excitation at 561nm) were demonstrated, in which ratio of acidic/neutral signals were significantly changed in a pH-dependent manner, particularly in the range between pH5 and pH8 (S3 Fig). Based on these data, mKeima-LC3B could monitor a wide range of the maturation step of autophago-somes into autolysosomes. Further, it was speculated that the pH value of mKeima-LC3B-con-taining compartments under nutrient-rich conditions in living cells ranged from pH6 to pH7, while those under starvation conditions were around pH5 (S3 Fig). Together, mKeima-LC3B fusion protein properly shows pH-dependent fluorescence signals as previously reported [20].

To determine whether mKeima-LC3B-based monitoring for the autophagy-endolysosomal system was compatible with high-throughput approaches, we analyzed mKeima-LC3B-expressing MEFs using a flow cytometer. A clear shift in acidic/neutral ratio of detected clus-ters were observed in live-cell analysis, in which neutral signal-dominated (Q1; 0.1%, Q3; 85.5%) and acidic signal-dominated (Q1; 15.9%, Q3; 8.3%) distributions were detected under autophagy-suppressed (DMEM+CQ) and starved (EBSS) conditions, respectively, when com-pared to normal nutrient-rich conditions (DMEM) (Q1; 4.9%, Q3; 38.1%) (Fig 6B–6D). These results demonstrate that flow cytometry-based live-cell analysis of mKeima-LC3B can be appli-cable to high-throughput screening approaches. Collectively, in contrast to *in vivo* (Fig 4),

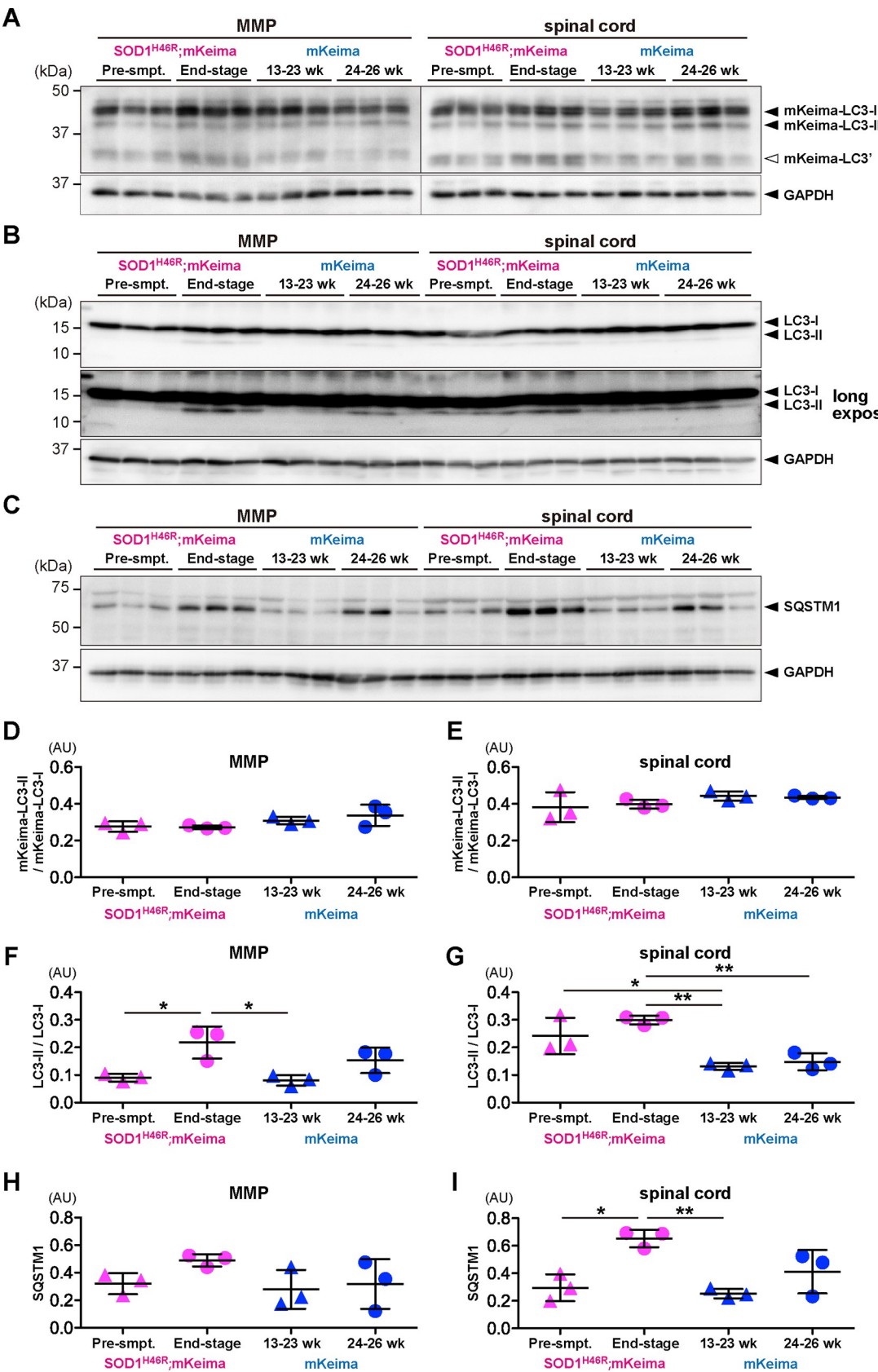

**Fig 4. Effects of mutant SOD1 overexpression on autophagy-related proteins *in vivo*.** (A-C) Western blot analysis of mKeima-LC3B, endogenous LC3 and SQSTM1 in double transgenic [SOD1$^{H46R}$;mKeima (KLC3_44)] and a single mKeima-LC3B transgenic (mKeima) mice (KLC3_44). Three pre-symptomatic mice (Pre-smpt.; 13–23 weeks of age) and three end-stage mice (24–26 weeks of age) and their age-matched mKeima-LC3B transgenic mice were used. Ten μg of total proteins from each tissue [MMP (midbrain + medulla + pons) and spinal cord] was subjected to SDS-PAGE and analyzed by immunoblotting using anti-LC3 and anti-SQSTM1 (p62) antibodies. Western blots for (A) mKeima-LC3B, (B) endogenous LC3 and (C) SQSTM1 are shown. White arrowhead shown on the right of panel (A) indicates a possible truncated mKeima-LC3B (indicated as mKeima-LC3'). GAPDH was used for a loading control. Positions of size-markers are shown on the left. (D-I) Quantification of signal intensities in western blots. Ratios (in arbitrary unit; AU) of (D) mKeima-LC3-II/mKeima-LC3-I/GAPDH in MMP, (E) mKeima-LC3-II/mKeima-LC3-I/GAPDH in the spinal cord, (F) endogenous LC3-II/LC3I/GAPDH in MMP, (G) endogenous LC3-II/LC3I/GAPDH in the spinal cord, (H) SQSTM1/GAPDH in MMP and (I) SQSTM1/GAPDH in the spinal cord are shown. Values are expressed as mean ± (S.D.). Individual data points are also shown. Statistical significance was evaluated by one-way ANOVA with Bonferroni's post hoc test; *$p < 0.05$, **$p < 0.01$.

mKeima-LC3B must be a useful molecular reporter to monitor the autophagy-endolysosomal system under cultured conditions *in vitro*.

## Live-cell imaging of mKeima-LC3B-expressing MEFs under nutrient-rich and starvation conditions

To clarify the intracellular behavior of mKeima-LC3B under nutrient-rich and starvation conditions, we conducted a live-cell imaging of mKeima-LC3B in mKeima-LC3B-expressing MEFs. We adopted an experimental setting in which cultured medium was repeatedly replaced from DMEM (nutrient-rich) to EBSS (starvation), and then from EBSS back to DMEM. First, we confirmed that medium change process itself did not affect the acidic/neutral signal ratio of mKeima-LC3B-positive compartments (S1 Movie and S2 Movie). Surprisingly, when cultured medium was repeatedly switched, acidic/neutral signal ratios were rapidly and reversibly changed, indicating that pH in mKeima-LC3B-localizing compartments was dynamically interchangeable between neutral and acidic ranges (Fig 7A, Fig 8A, S3 Movie and S4 Movie).

To confirm whether the fluctuations observed in mKeima-LC3B-positive signals actually reflected the changes in the intralumenal pH, we performed a simultaneous staining with acidic vesicles with LysoTracker. As expected, a majority of perinuclear LysoTracker-positive vesicles representing late-endosomes, amphisomes and/or autolysosomes were also mKeima-

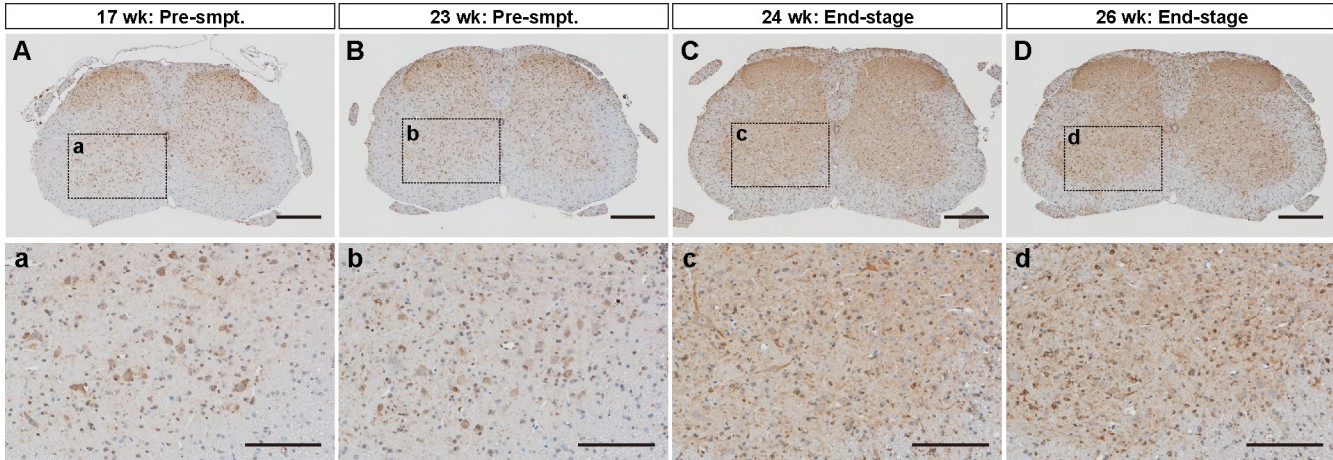

**Fig 5. Immunohistochemical analysis of LC3 in double transgenic mice expressing mutant SOD1 and mKeima-LC3B.** Lumbar spinal cord (L4-L5) tissues, which was prepared from (A, a, B, and b) two pre-symptomatic and (C, c, D and d) two end-stage double transgenic mice [SOD1$^{H46R}$;mKeima (KLC3_44)] expressing both mutant SOD1 (SOD1$^{H46R}$) and mKeima-LC3B, were immunostained with anti-LC3 antibody. Scale bars indicate (A-D) 200 μm and (a-d) 100 μm.

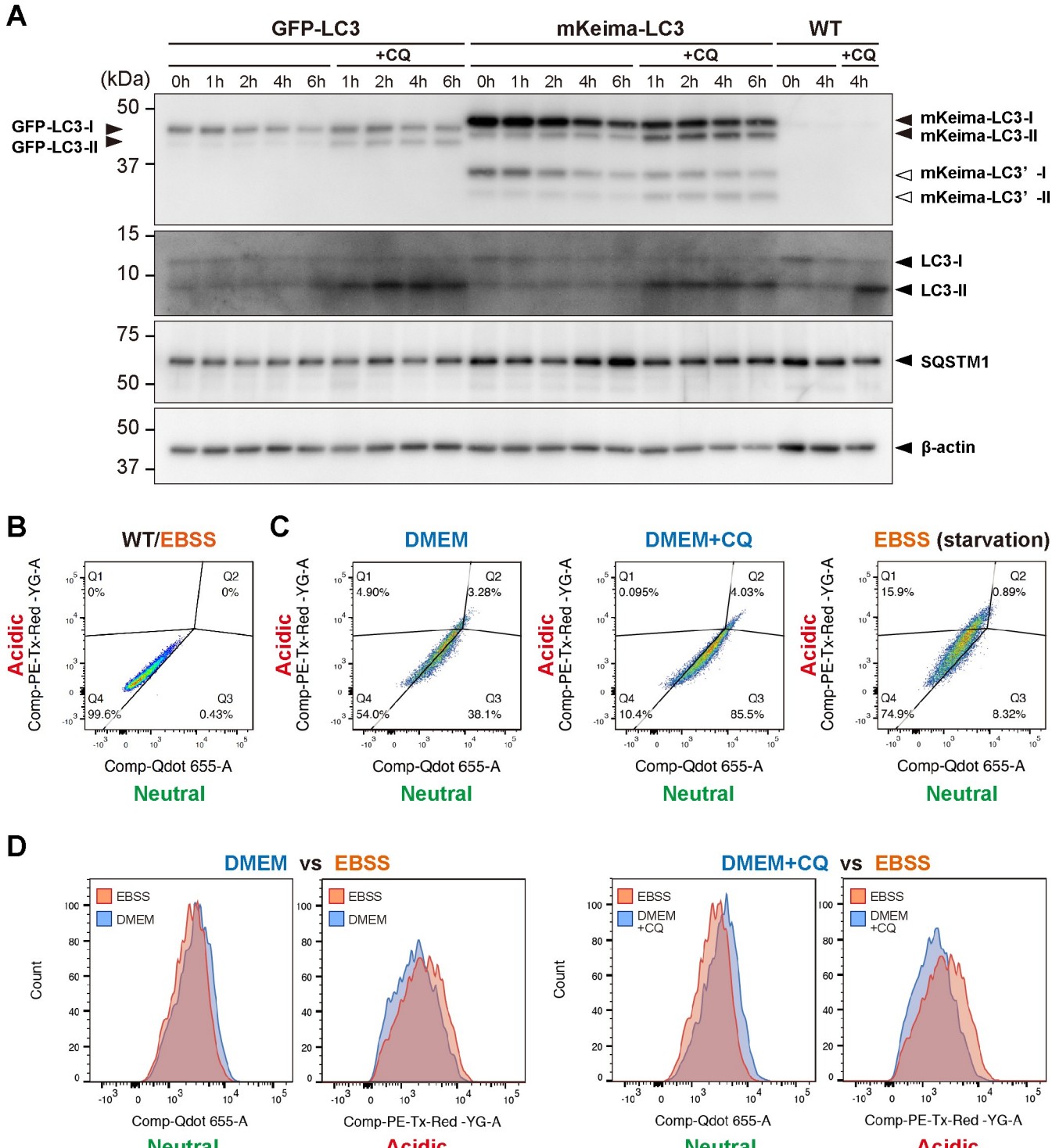

**Fig 6. Effects of starvation on mKeima-LC3B fusion protein in MEFs.** (A) Western blot analysis of GFP-LC3 and mKeima-LC3B (top panel), endogenous LC3 (2nd panel) and SQSTM1 (3rd panel). MEFs prepared from GFP-LC3-tg, mKeima-LC3B-tg (BDKLC3_17–1) and wild-type mice were cultured in DMEM followed by incubation in EBSS with or without 50 µM chloroquine (CQ) for indicated time-periods. Two-point three (2.3) µg of total proteins from each sample was subjected to SDS-PAGE and analyzed by immunoblotting using anti-LC3 and anti-SQSTM1 antibodies. β-actin was used for a loading control. Positions of GFP-LC3-I and -LC3-II are shown on the left. Positions of mKeima-LC3B-I and -LC3-II, endogenous LC3-I and LC3-II, and SQSTM1 are shown on the right. White arrowheads shown on the right indicate a possible truncated mKeima-LC3B (indicated as mKeima-LC3'). Size-markers are shown on the left. (B) Flow cytometric analysis of the fluorescent signals in wild-type (WT) MEFs treated with EBSS. Four quadrants (Q1-Q4) wereassigned based on the distribution of background signals observed in WT MEFs, in which Q4 contained the maximum number of cells within its minimum area. (C) Flow

cytometric analysis of the mKeima-derived fluorescent signals in mKeima-LC3B-expressing MEFs. Data of MEFs treated with DMEM, DMEM + 50 μM CQ and EBSS (starvation) are shown. Horizontal and vertical axes indicate the emission signal intensities of mKeima at neutral (Comp-Qdot 655-A) and acidic (Comp-PE-Tx-Red-YG-A) pH, respectively. (D) Comparative analysis of the mKeima-derived fluorescent signals obtained from panel B.

LC3 positive. Further, they similarly showed pH-dependent fluorescence signals upon changes in cultured media (Fig 7B, S4 Fig, S5 Movie, S6 Movie and S7 Movie). Notably, under prolonged starvation conditions; i.e., 60 to 120 min in EBSS, acidic signals in mKeima-LC3B-/LysoTracker-positive vesicle compartments were further increased (Fig 7B), which was fully consistent with previous findings [18]. These data suggest that a rapid and reversible change of pH in LC3-positive autophagosomes and/or amphisomes may precede the vesicle-fusion-based gradual acidification and maturation of autolysosomes.

Next, to further confirm such notions, we conducted a live-cell imaging using a high-resolution Airyscan analysis and investigated the more detailed behavior of mKeima-LC3B-positive vesicles. Under nutrient-rich conditions (DMEM), acidic/neutral ratio of fluorescence signals in mKeima-LC3B-positive vesicles was unchanged (S8 Movie). Under repeated-exchanging conditions between nutrient-rich and starvation, acidic/neutral signal ratio was rapidly and reversibly changed, in which the signal shifts of the inner-side of vesicular membrane were more evident than those of the cytoplasmic face (Fig 7C, Fig 8B and S9 Movie). Notably, such changes seemed to occur autonomously without committing the vesicular fusion (Fig 7C, Fig 8B and S9 Movie). Quantification of these signal shifts using a high-content analysis revealed that increased signal ratio by starvation reached maximum level within ~20 sec after challenging starvation medium (Fig 8C and S5 Fig). Further, such acidic signals were reversed within ~10 sec when medium returned to nutrition-rich one (Fig 8C). These results indicate that intraluminal pH of mKeima-LC3B-positive vesicles is rapidly changeable upon nutritional conditions of culture media.

Finally, to test whether these rapid and reversible changes in intraluminal pH of mKeima-LC3B-positive vesicles were associated with genuine maturation of autophagosomes and/or endosomes, we conducted a live-cell imaging as well as quantification of acidic/neutral signals for mKeima-LC3B in the presence or absence of CQ. Treatment with CQ resulted in a gradual decrease of acidic/neutral signal ratio, whose levels were reached to minimum over ~ 6 min of CQ treatment (Fig 9A and S10 Movie), indicating that deacidification by CQ proceeded much slower than did by medium switch from EBSS to DMEM. Interestingly, when the medium was changed from DMEM to EBSS+CQ, prior to CQ-dependent deacidification, a rapid and transient acidification was still observed (Fig 9B and S11 Movie). Fully-acidified mKeima-LC3B-positive vesicles by EBSS were also similarly deacidified by CQ (Fig 9C, Fig 9D, S12 Movie and S13 Movie). Further, vesicles under fully-deacidified conditions by CQ were rapidly reacidified by EBSS treatment (Fig 9E, Fig 9F, S14 Movie and S15 Movie). Notably, MEFs with persistent exposure to CQ were totally unresponsive to the medium switch from DMEM to EBSS; i.e., from DMEM+CQ to EBSS+CQ (S16 Movie). These results suggest that a rapidly changeable intraluminal pH of mKeima-LC3B-positive vesicles depends, at least in part, on the process of the authentic vacuolar-type H+-ATPase (V-ATPase)-linked maturation system of autophagosomes and/or endosomes [27].

## Discussion

We here established a number of tg mouse lines that were expressing the mKeima-LC3B chimeric protein as a possible novel means for monitoring the autophagy-endolysosomal system. By conducting a series of *in vivo* as well as *in vitro* experiments, we revealed that mKeima-LC3B could be a sensitive reporter molecule for monitoring the autophagy-endolysosomal

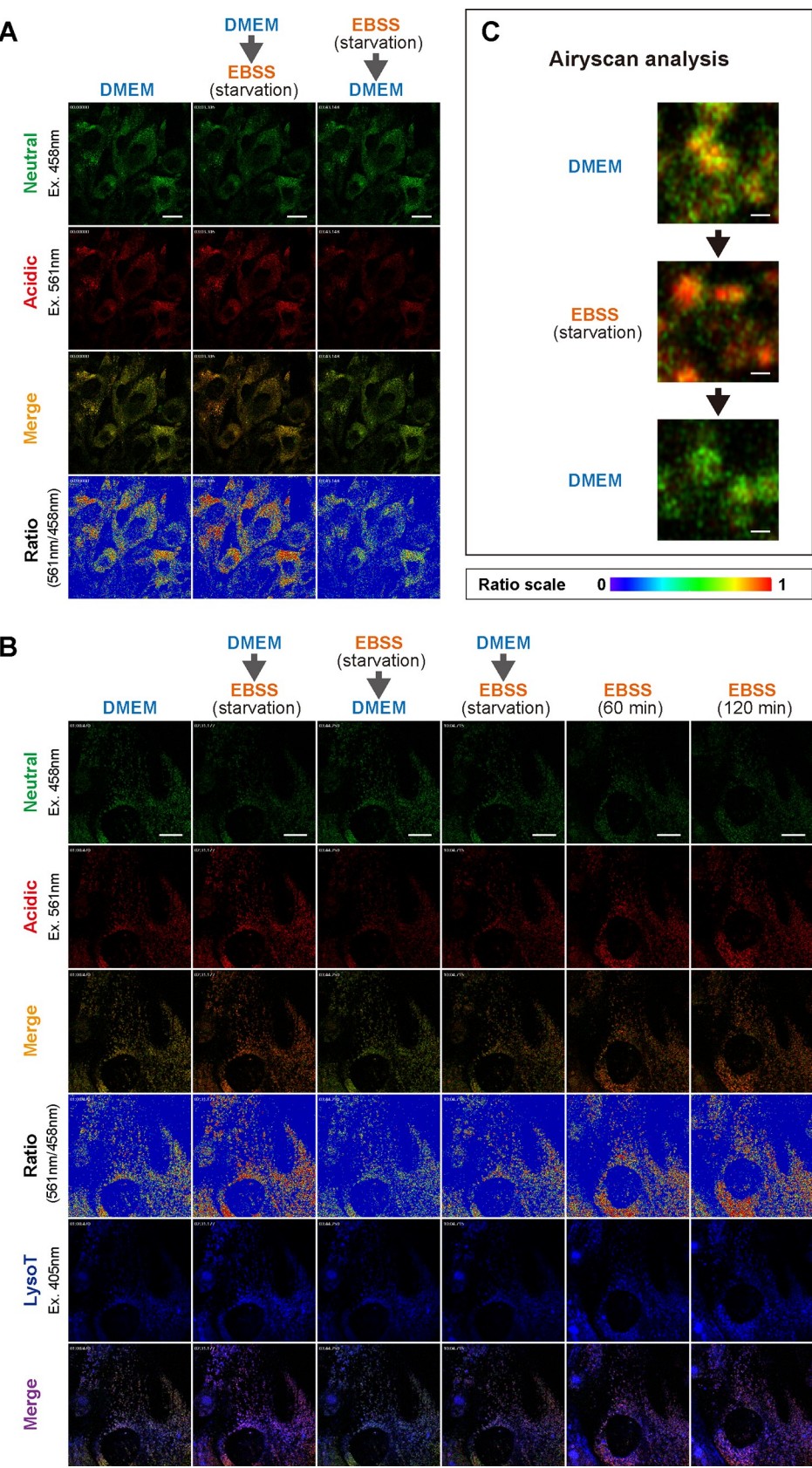

**Fig 7. Effects of starvation on mKeima-LC3B-positive vesicular compartments in MEFs.** (A) Representative images of mKeima-LC3B-expressing MEFs. MEFs prepared from mKeima-LC3B-tg mice (BDKLC3_17–1) were cultured in DMEM. After incubation in DMEM for 2 min, medium was repeatedly changed from DMEM to EBSS and from EBSS to DMEM every 80 sec. Images in each row represent as follows: upper row; Neutral (mKeima; ex. 458 nm, green), 2nd row; Acidic (mKeima; ex. 561 nm, red), 3rd row; Merge (mKeima; Neutral + Acidic) and lower row; ratio [mKeima; 561 nm (Acidic)/458 nm (Neutral)]. Scale bars indicate 40 μm. (B) Representative images of mKeima-LC3B and LysoTracker double-positive vesicles. MEFs prepared from mKeima-LC3B-tg mice were cultured in DMEM. After incubation in DMEM for 1 min, medium was repeatedly changed from DMEM to EBSS and from EBSS to DMEM every 80 sec. Images in each row represent as follows: upper row; Neutral (mKeima; ex. 458 nm, green), 2nd row; Acidic (mKeima; ex. 561 nm, red), 3rd row; Merge (mKeima; Neutral + Acidic), 4th row; ratio [mKeima; 561 nm (Acidic)/458 nm (Neutral)], 5th row; LysoT (LysoTracker blue; ex. 405 nm, blue) and lower row; Merge [mKeima (Neutral + Acidic) + LysoTracker blue]. Scale bars indicate 20 μm. (C) Representative high-resolution images of mKeima-LC3B-positive vesicles by Airyscan analysis. Merged images (mKeima; Neutral + Acidic) are shown. Scale bars indicate 0.5 μm.

system under *in vitro* cultured conditions. However, the use of mKeima-LC3B-tg mice may not be appropriate due to an aggregate-prone nature of overexpressed mKeima-LC3B *in vivo*, particularly in the brain.

Since dysfunction of the autophagy-endolysosomal system is thought to be associated with many pathological conditions such as cancer, inflammation and neurodegenerative diseases, it must be very important to properly and precisely understand each step of the autophagy-endolyosomal system, which includes the autophagosome formation, maturation and degradation. Thus far, LC3 tagged with a green fluorescent protein, GFP-LC3, has mostly been utilized to detect the formation of autophagosomes not only in cells but also at an *in vivo* level [9]. Nonetheless, there are still some shortcomings to monitor the flux throughout the entire autophagic system using this fusion protein due to its vulnerability to acidic conditions. To overcome such weaknesses, a number of improved molecular probes, such as RFP-GFP-LC3 [12–14], mCherry-EGFP-LC3 [15], GFP-LC3-RFP-LC3ΔG [16], mTagRFP-mWasabi-LC3 [17] and pHluorin-mKate2-tagged LC3 [18], has been developed thus far. In this study, we newly developed an alternative marker; i.e., mKeima-LC3B, for monitoring the autophagy-endolysosomal activity, and investigated its usefulness in *in vivo* animal as well as *in vitro* cell culture studies.

Previously, it has been shown that GFP-LC3-expressing tg mice is a useful tool to study autophagic responses to starvation *in vivo* [9]. We and others have also demonstrated that GFP-LC3 emerges as a sensitive molecular marker to monitor the progression of disease in an ALS mouse model *in vivo* [10, 11]. Therefore, to prove as to whether mKeima-LC3B could response to stresses including food deprivation and neurodegeneration, we conducted a series of *in vivo* studies. Unexpectedly, our results clearly demonstrated that unlike endogenous LC3 or GFP-LC3, mKeima-LC3B did not properly respond to these stresses, even though autophagy itself was dysregulated in corresponding mice. Notably, extended expression of mKeima-LC3B rather seemed to result in accelerated proteolytic stresses, particularly, in the CNS. Since the mKeima protein may have some intrinsically aggregate-prone properties conferring resistance to acidic conditions [20], it is speculated that mKeima-LC3 is less-efficiently degraded in autolysosomes and/or lysosomes as previously reported [28]. Taken together, mKeima-fused LC3-expressing animals may not be an appropriate tool to monitor the autophagy-endolysosomal system under stressed conditions *in vivo*.

Contrary to *in vivo* experiments, we were able to show that mKeima-LC3B could be utilized to sensitively monitor the dynamic changes of maturation states in the autophagy-endolysosomal system in cultured cells. It has been reported that pH value of the cytosol ranges between 7.2 and 7.6 [29, 30]. It is also estimated that the intralumenal pH of nascent autophagosomes is close to that of the cytosol. On the other hand, the intralumenal pH of other organelle and vesicles are varied, in which the endoplasmic reticulum, cis-Golgi, trans-Golgi, secretory vesicles,

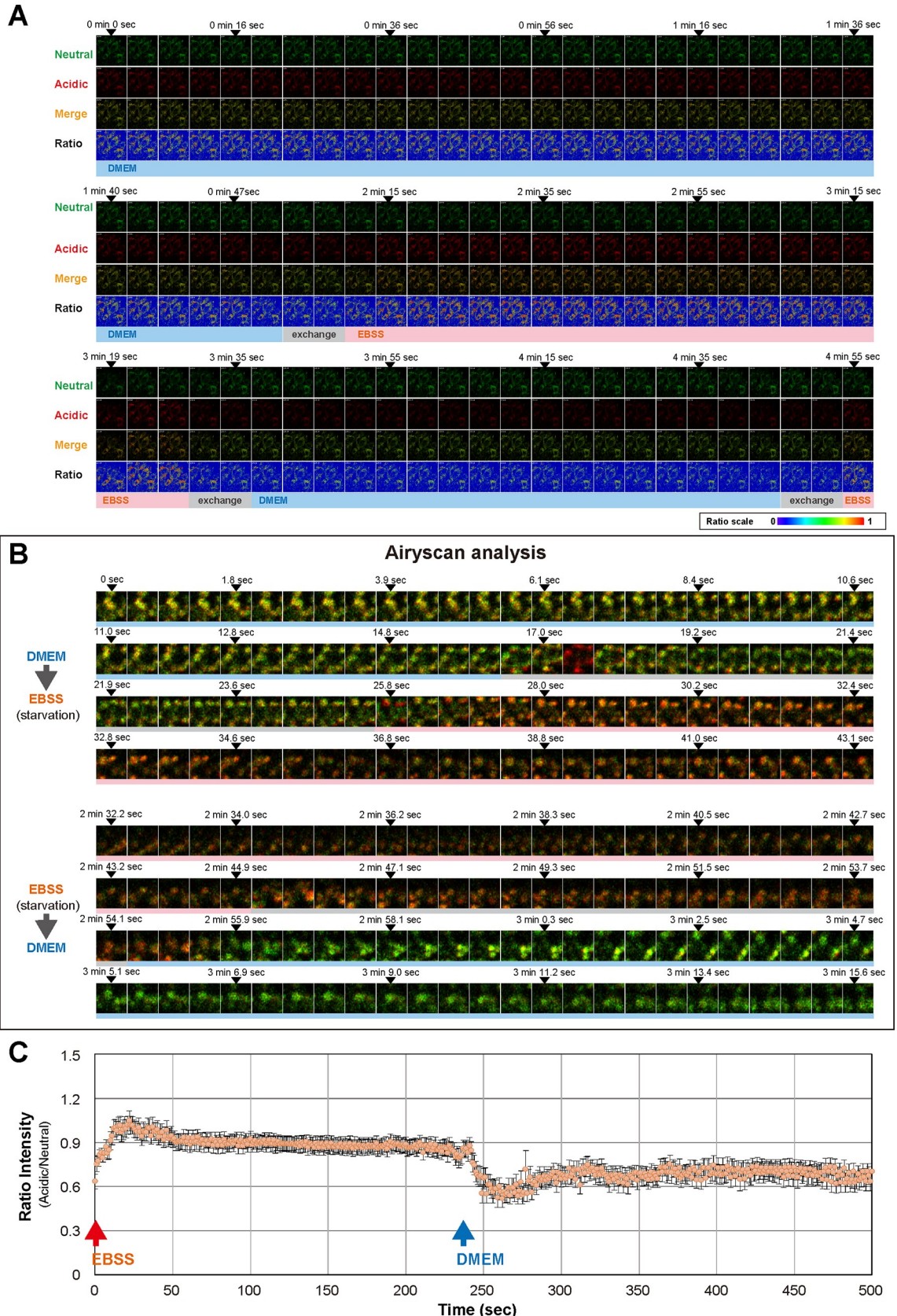

**Fig 8. Live-cell analysis of MEFs expressing mKeima-LC3B.** (A) Representative live-cell time-lapse images of MEFs expressing mKeima-LC3B (related to Fig 7A). MEFs prepared from mKeima-LC3B-tg mice (BDKLC3_17–1) were cultured in DMEM. After incubation in DMEM for 2 min, medium was repeatedly changed from DMEM to EBSS and from EBSS to DMEM every 80 sec. Images (Z-stack = 2) were captured every 4 sec. Images in each row represent as follows: upper row; Neutral (mKeima; ex. 458 nm, green), 2nd row; Acidic (mKeima; ex. 561 nm, red), 3rd row; Merge (mKeima; Neutral + Acidic) and lower row; ratio [mKeima; 561 nm (Acidic)/458 nm (Neutral)]. Lap-times and medium conditions are shown in the top and bottom, respectively. (B) Representative high-resolution live-cell time-lapse images of mKeima-LC3B-positive vesicles under transitional conditions from nutrient-rich (DMEM) to starvation (EBSS) and from starvation (EBSS) to nutrient-rich (DMEM) states (related to Fig 7C). Images were captured every 300 msec. Lap-times are shown in the top. Blue and pink bars shown in the bottom indicate the incubation period with DMEM and EBSS, respectively. Gray bar indicates the period of medium exchange. (C) Quantitative analysis of changes in mKeima-derived fluorescent signal ratio (Acidic/Neutral) under transitional conditions from nutrient-rich (DMEM) to starvation (EBSS) and from starvation (EBSS) to nutrient-rich (DMEM) states. Data were captured for a total of 500 sec at 1 sec intervals [observed field; n = 39 (total number of cells; n ≈ 480)]. Values are expressed as mean ± (S.E.M.).

early endosomes, recycling endosomes, late endosomes and lysosomes are estimated to be ~7.2, ~6.7, ~6.0, 5.2–5.7, ~6.3, ~6.5, ~6.0 and ~5.5, respectively [31]. Since ratio of the bimodal excitation fluorescent spectrum (acidic/neutral) of mKeima-LC3B allowed to determine the approximate pH values ranging between pH5 and pH8, estimating the maturation states of mKeima-LC3B-resided compartments, especially autophagosomes, endosomes and lysosomes, in cells might be possible. Indeed, in this study, we can estimate that the pH value of mKeima-LC3B-containing compartments in living MEFs under nutrient-rich conditions ranges from pH6 to pH7, while those under starvation conditions are around pH5.

One of the remarkable findings in this study is that although the full maturation of autophagosomes by the fusion with lysosomes seems to be slow (in a range of hours) as has previously been reported [18], changes of pH in mKeima-LC3B-positive vesicles is rather very rapid (in a range of seconds). Furthermore, to our surprise, the intraluminal pH of mKeima-LC3B-positive vesicles is reversibly changeable upon nutritional conditions of culture media. Importantly, these shifts are completely suppressed by the CQ treatment. Thus, a rapid and reversible change of pH in mKeima-LC3-positive autophagosomes observed in this study is likely to rely on the function of genuine vesicular proton pumps, which may precede the vesicle-fusion-based gradual acidification and ultimate irreversible maturation of autolysosomes.

It has been well studies on the relationship between amino acid starvation and the activation of autophagy [1]. Amino acid starvation promotes the formation of unc-51-like kinase 1 (ULK1)-/LC3-/SQSTM1-positive nascent autophagosomes, which is regulated by the mechanistic target of rapamycin (mTOR) complex 1 (mTORC1)-mediated sensing of amino acids. Resulting autophagosomes sequentially mature through multiple steps comprising of vesicle fusion with endosomes (to form amphisomes) and/or lysosomes (to form autolysosomes). Over such maturation steps, the vesicular proton pump V-ATPase, which is localized not only on lysosomal membrane but also on most of endocytic vesicles, plays an essential role in the intralumenal acidification [27]. Activity of V-ATPase is regulated by many different mechanisms, which includes the association of the catalytic $V_1$ domain with the membrane-associated $V_0$ domain, the coupling efficiency between ATP hydrolysis and proton translocation, and the interaction with specific lipid environments [32]. Recently, it has been revealed that a reversible assemble of V-ATPase; i.e., the association and dissociation of $V_1$ and $V_0$, in response to the changes in amino acid levels is linked to the luminal pH and the catalytic activity of V-ATPase, but not to the phosphatidylinositol-3 kinase (PI3K) and mTORC1 activities [33]. Thus, since autophagosome-lysosome fusion itself is also independent of the V-ATPase-mediated acidification [34], it is highly likely that the activation of V-ATPase prior to lysosome fusion leads to rapid acidification of mKeima-LC3B-positive vesicles. However, at present, the exact molecular pathway, which is responsible for the amino acid starvation-induced

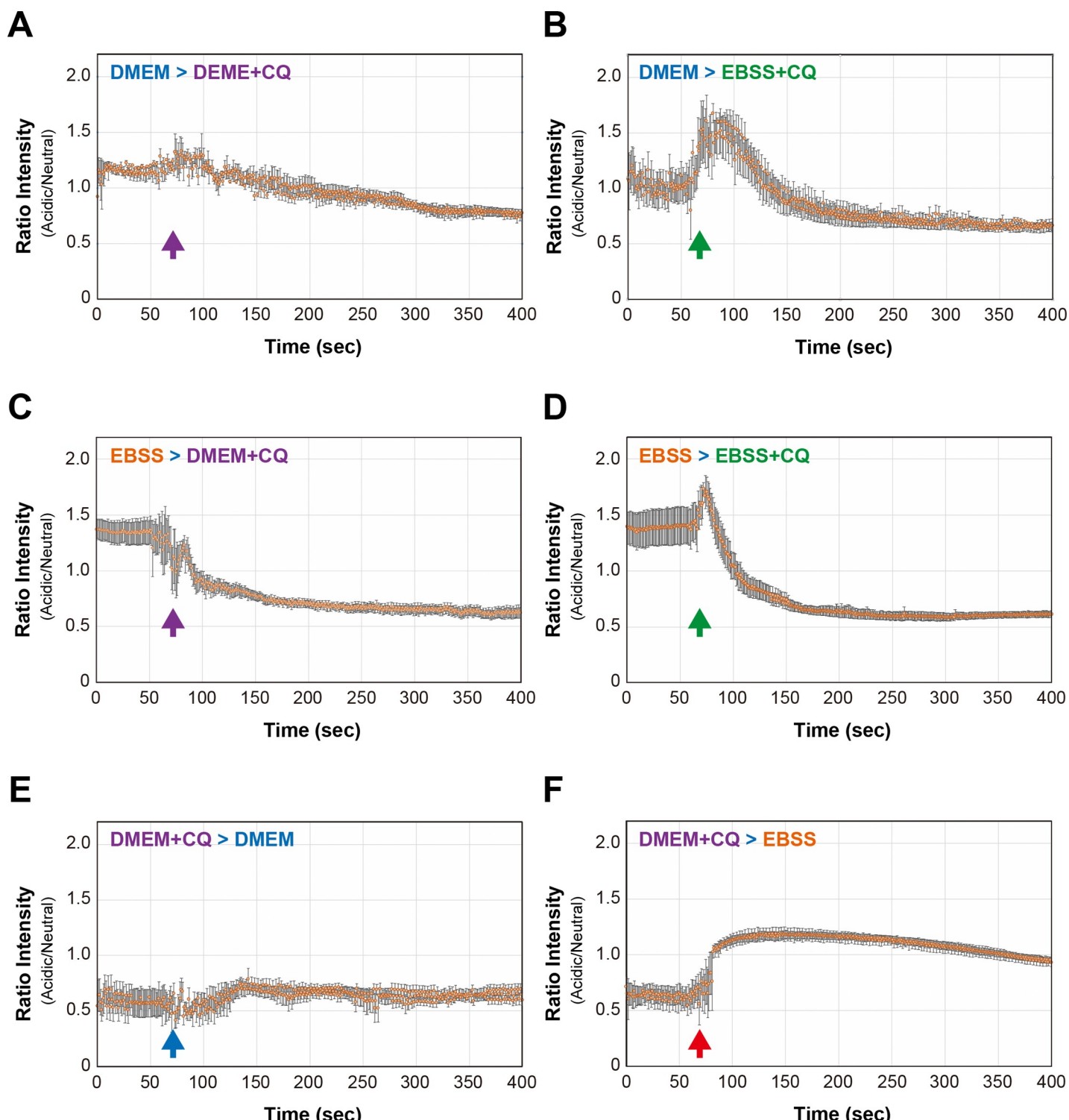

**Fig 9. Quantitative analysis of changes in mKeima-derived fluorescent signal ratio in the presence of chloroquine.** Changes in mKeima-derived fluorescent signal ratio (Acidic/Neutral) under transitional conditions from (A) nutrient-rich (DMEM) to DMEM + 50 μM chloroquine (CQ) (observed field; n = 5, total number of cells; n ≈ 60), (B) DMEM to starvation (EBSS) +CQ (observed field; n = 4, total number of cells; n ≈ 50), (C) EBSS to DMEM + CQ (observed field; n = 6, total number of cells; n ≈ 70), (D) EBSS to EBSS + CQ (observed field; n = 4, total number of cells; n ≈ 50), (E) DMEM + CQ to DMEM (observed field; n = 7, total number of cells; n ≈ 70) and (F) DMEM + CQ to EBSS (observed field; n = 5, total number of cells; n ≈ 60) states are shown. Values are expressed as mean ± (S.E.M.).

activation of V-ATPase and thus rapid acidification (~ within 20 sec) of these vesicles, remains unclear.

One perplexing but important question arising from this study includes as to how a rapid deacidification (~ within 10 sec) of mKeima-LC3B-resided compartments occurs by changing the media from EBSS back to DMEM. When we added CQ to cell cultures, the deacidification of mKeima-LC3B-positive vesicles gradually and directionally proceeded from the peripheral to center of the cell over ~6 min (see Fig 9D and S13 Movie), indicative of a slow penetration of CQ into the cells. Conversely, it is reasonable to assume that rapid and synchronized deacidification of mKeima-LC3B-positive vesicles is rather mediated by virtue of a certain signaling pathway to sense the amino acid repletion in culture media. Thus far, besides V-ATPase, the alkali cation/proton exchanger SLC9A6/NHE6, whose mutations are linked to Christianson syndrome, is known to be implicated in the intraluminal pH homeostasis in endocytic vesicles, namely in recycling endosomes [35]. However, to the best of our knowledge, there have been no studies demonstrating the active-deacidification of autophagosomes and/or endocytic vesicles. On the other hand, it has been reported that the activity of V-ATPase itself in the endocytic pathway is reversibly regulated [32]. Interestingly, the intracellular concentration of calcium ions was also reversibly changed by responding to the presence and/or absence of amino acids in culture media, in which the addition and withdrawal of amino acids result in decrease and increase of intracellular calcium concentrations, respectively [36]. Although the authors in this interesting study have claimed that increased calcium ions activate $Ca^{2+}$/calmodulin-dependent kinase kinase (CaMKK) and AMP-activated protein kinase (AMPK) with concomitant inhibition of mTORC1, thereby inducing autophagy (in a range of hours), the response of decrease in the intracellular calcium concentration to amino acid supplement was, in fact, remarkably rapid (in a range of seconds) [36] as we observed in DMEM-induced rapid deacidification. It is also notable that the vesicular pH (pH6-7) is further shifted towards alkaline pH (~pH8) when CQ is added to DMEM (see Fig 9A and S10 Movie), indicating that the moderate but persistent activation of V-ATPase occurs to maintain the physiological pH (pH6-7) in autophagosomes and/or endocytic vesicles even under nutrition-rich conditions.

Based on these notions, we hypothesize two possible mechanisms to explain the rapid deacidification of mKeima-LC3B-resided compartments by changing culture media from EBSS to DMEM. First, amino acid repletion can trigger a rapid decline of the V-ATPase activity via certain signaling pathways; e.g., calcium ion-mediated signaling. Consequently, the ability to maintain an intraluminal acidity is instantly lost, resulting in autophagosomes and/or endocytic vesicles with homeostatic neutral pH (pH6-7) within ~10 sec. Second, there may exist a yet-to-be-identified active deacidification mechanism that responds to extracellular amino acids. However, at this stage, we could not fully explain the mechanism and/or signaling pathway for these rapid-reversible phenomena in an evidence-based manner.

Another issue that has yet to be addressed is the exact and/or precise definition of the mKeima-LC3B-labeled compartments. It has been well described that endocytic pathway and autophagic pathway are interrelated [37, 38]. In addition, several recent studies have demonstrated that LC3-positive vesicles include not only autophagosomes and amphisomes [21] but also endosomes derived through distinct endocytic pathways, such as LC3-associated phagocytosis (LAP) [39], LC3-associated endocytosis (LANDO) [40] and LC3-conjugated multivesicular bodies containing extracellular vesicles to be secreted [41]. Thus, it is possible that mKeima-LC3B-positive vesicles include other than autophagosomes and/or amphisomes. To precisely and distinctly define mKeima-LC3B-positive compartments, development an additional set of specific vesicular markers with fluorophore, which can be simultaneously used with mKeima, will be required.

In conclusions, the intralumenal pH of mKeima-LC3B-residing compartments, probably autophagosomes, endosomes and autolysosomes, can be rapidly and reversibly changed upon nutritional conditions. Currently, it has been believed that the maturation of autophagosomes toward autolysosomes and/or lysosomes is the one-way directional pathway [1, 6, 7]. However, it is possible that at some stages of maturation, autophagosomes and/or endosomes still place in undetermined fate before tilting toward the irreversible maturation step of vesicular compartments. Currently, the physiological meaning of such reversible rapid changes of the intra-lumenal pH is unclear. In order to answer these questions, further studies will be required. Detailed characterization of autophagosome maturation step by using mKeima-LC3B as a molecular probe will give us more clues to understanding the autophagy-endolysosomal process in the manifestation of many physiological as well as pathological conditions.

## Supporting information

**S1 Fig. Schematic diagram of transgene construct for mKeima-LC3B transgenic mouse.** Transgene construct consists of cytomegalovirus enhancer (E), chicken β-actin promoter (Pro), rabbit β-globin splice acceptor (S), mKeima-Red cDNA, full-length human MAP1LC3B cDNA (LC3B), and rabbit β-globin poly A (Poly A). Positions of restriction enzymes used for cloning and primers for genotyping are shown.
(TIF)

**S2 Fig. Immunohistochemical analysis of LC3 in tissues from wild-type and mKeima-LC3B transgenic mice under starvation conditions.** (A) Cerebral cortex, (B) hippocampus, (C) cerebellum, (D) pons (facial nucleus), and (E) lumbar spinal cord (L4-L5), which were pre-pared from wild-type and mKeima-LC3B transgenic (tg) mice under either fed (control) or starved (48 h) conditions, were fixed and immunostained with anti-LC3 (α-LC3) or anti-mKeima (α-mKeima) antibody. Mouse lines used in this analysis were (A-D) BDKLC3_17–1 and (E) KLC3_44. Scale bars indicate (A, a-f) 200 μm, (A, g-l) 100 μm, (B, a-f) 200 μm, (B, g-l) 100 μm, (C, a-f) 200 μm, (C, g-l) 100 μm, (D, a-f)) 200 μm, (D, g-l) 100 μm, and (E) 200 μm.
(TIF)

**S3 Fig. The pH sensitivity of mKeima-LC3B.** After fixation, fluorescent signals for mKeima in mKeima-LC3B-expressing MEFs, which were prepared from mKeima-LC3B-tg mice (BDKLC3_17–1), were detected (Pre). Then, fixed cells were buffered at (A) pH 4.0, (B) pH 5.0, (C) pH 6.0, (D) pH 7.0, (E) pH 8.0, or (F) pH 9.0, or treated with (G) EBSS (pH 7.4–7.6), followed by detection of fluorescent signals (Post). (H) For comparison, images for living cells treated with DMEM or EBSS are shown. Scale bars indicate 50 μm.
(TIF)

**S4 Fig. Representative live-cell time-lapse images of MEFs expressing mKeima-LC3B.** MEFs prepared from mKeima-LC3B tg mice (BDKLC3_17–1) were cultured in DMEM. After incubation in DMEM for 1 min, medium was repeatedly changed from DMEM to EBSS and from EBSS to DMEM every 80 sec. Images (Z-stack = 1) were captured every 2.5 sec. Images in each row represent as follows: upper row; Neutral (mKeima; ex. 458 nm, green), 2nd row; Acidic (mKeima; ex. 561 nm, red), 3rd row; Merge (mKeima; neutral + acidic), 4th row; ratio [mKeima; 561 nm (acidic)/458 nm (neutral)], 5th row; LysoT (LysoTracker blue; ex. 405 nm, blue), and lower row; Merge [mKeima (neutral + acidic) + LysoTracker blue]. Lap-times and medium conditions are shown in the top and bottom, respectively. As a negative control, images of wild-type MEFs cultured in DMEM are also shown. Scale bar indicates 20 μm.
(TIF)

**S5 Fig. Quantitative analysis of changes in mKeima-derived fluorescent signal ratio (related to Fig 8).** Changes in mKeima-derived fluorescent signal ratio (Acidic/Neutral) under transitional conditions from nutrient-rich (DMEM) to starvation (EBSS) states are shown. Two independent experiments, in which data were captured for 100 sec at 1 sec intervals [EBSS 1st (magenta circle); observed field; n = 29 (total number of cells; n ≈ 350) and EBSS 2nd (yellow circle); observed field; n = 55 (total number of cells; n ≈ 670)], were performed. Signal ratio under nutrient-rich conditions [DMEM (blue circle); observed field; n = 50 (total number of cells; n ≈ 610)] was also monitored as a control. Values are expressed as mean ± (S.E.M.).
(TIF)

**S1 Movie. A time-lapse movie of MEFs expressing mKeima-LC3B under conditions with repeated changes in nutrient-rich medium (DMEM).** After incubation in DMEM for 60 min, medium was changed from DMEM to fresh DMEM. Merged images (Z-stack = 2) for mKeima signals [neutral (green) + acidic (red)] were captured every 5 sec. Scale bar indicates 20 μm.
(MOV)

**S2 Movie. A time-lapse movie of MEFs expressing mKeima-LC3B under conditions with repeated changes in starvation medium (EBSS).** After incubation in EBSS for 60 min, medium was changed from EBSS to fresh EBSS. Merged images (Z-stack = 2) for mKeima signals [neutral (green) + acidic (red)] were captured every 5 sec. Scale bar indicates 20 μm.
(MOV)

**S3 Movie. A time-lapse movie of MEFs expressing mKeima-LC3B under nutrient-rich conditions (DMEM) (related to Figs 7 and 8).** Merged images (Z-stack = 2) for mKeima signals [neutral (green) + acidic (red)] were captured every 4 sec. Scale bar indicates 20 μm.
(MOV)

**S4 Movie. A time-lapse movie of MEFs expressing mKeima-LC3B under conditions with repeated changes in media between nutrient-rich (DMEM) and starvation (EBSS) states (related to Figs 7 and 8).** After incubation in DMEM for 2 min, medium was repeatedly changed from DMEM to EBSS and from EBSS to DMEM every 80 sec. Merged images (Z-stack = 2) for mKeima signals [neutral (green) + acidic (red)] were captured every 4 sec. Scale bar indicates 20 μm.
(MOV)

**S5 Movie. A time-lapse movie of MEFs expressing mKeima-LC3B under conditions with repeated changes in media between nutrient-rich (DMEM) and starvation (EBSS) states (related to Fig 7).** After incubation in DMEM for 1 min, medium was repeatedly changed from DMEM to EBSS and from EBSS to DMEM every 80 sec. Images for LysoTracker blue (Z-stack = 1) (blue) were captured every 2.5 sec. Scale bar indicates 10 μm.
(MOV)

**S6 Movie. A time-lapse movie of MEFs expressing mKeima-LC3B under nutrient-rich conditions (DMEM) (related to Fig 7).** Merged images (Z-stack = 1) for signals of mKeima [neutral (green) and acidic (red)] and LysoTracker blue (blue) were captured every 2.5 sec. Scale bar indicates 10 μm.
(MOV)

**S7 Movie. A time-lapse movie of MEFs expressing mKeima-LC3B under conditions with repeated changes in media between nutrient-rich (DMEM) and starvation (EBSS) states**

**(related to Fig 7).** After incubation in DMEM for 1 min, medium was repeatedly changed from DMEM to EBSS and from EBSS to DMEM every 80 sec. Merged images (Z-stack = 1) for signals of mKeima [neutral (green) and acidic (red)] and LysoTracker blue (blue) were captured every 2.5 sec. Scale bar indicates 10 μm.
(MOV)

**S8 Movie. A time-lapse movie of mKeima-LC3B-positve vesicles under nutrient-rich conditions (DMEM) (related to Figs 7 and 8).** Merged images for mKeima signals [neutral (green) and acidic (red)] were captured every 300 msec. Scale bar indicates 0.5 μm.
(MOV)

**S9 Movie. A time-lapse movie of mKeima-LC3B-positve vesicles under conditions with repeated changes in media between nutrient-rich (DMEM) and starvation (EBSS) states (related to Figs 7 and 8).** Medium was repeatedly changed from DMEM to EBSS and from EBSS to DMEM every 80 sec. Merged images for mKeima signals [neutral (green) and acidic (red)] were captured every 300 msec. Scale bar indicates 0.5 μm.
(MOV)

**S10 Movie. A time-lapse movie of MEFs expressing mKeima-LC3B under transitional conditions from nutrient-rich (DMEM) to DMEM + 50 μM chloroquine (CQ) states (related to Fig 9A).** After incubation in DMEM for 80 sec, medium was changed from DMEM to DMEM + CQ. Merged images (Z-stack = 2) for mKeima signals [neutral (green) + acidic (red)] were captured every 5 sec. Scale bar indicates 20 μm.
(MOV)

**S11 Movie. A time-lapse movie of MEFs expressing mKeima-LC3B under transitional conditions from nutrient-rich (DMEM) to starvation (EBSS) + 50 μM chloroquine (CQ) states (related to Fig 9B).** After incubation in DMEM for 80 sec, medium was changed from DMEM to EBSS + CQ. Merged images (Z-stack = 2) for mKeima signals [neutral (green) + acidic (red)] were captured every 5 sec. Scale bar indicates 20 μm.
(MOV)

**S12 Movie. A time-lapse movie of MEFs expressing mKeima-LC3B under transitional conditions from starvation (EBSS) to nutrient-rich (DMEM) + 50 μM chloroquine (CQ) states (related to Fig 9C).** After incubation in EBSS for 60 min, medium was changed from EBSS to DMEM + CQ. Merged images (Z-stack = 2) for mKeima signals [neutral (green) + acidic (red)] were captured every 5 sec. Scale bar indicates 20 μm.
(MOV)

**S13 Movie. A time-lapse movie of MEFs expressing mKeima-LC3B under transitional conditions from starvation (EBSS) to EBSS + 50 μM chloroquine (CQ) states (related to Fig 9D).** After incubation in EBSS for 60 min, medium was changed from EBSS to EBSS + CQ. Merged images (Z-stack = 2) for mKeima signals [neutral (green) + acidic (red)] were captured every 5 sec. Scale bar indicates 20 μm.
(MOV)

**S14 Movie. A time-lapse movie of MEFs expressing mKeima-LC3B under transitional conditions from nutrient-rich (DMEM) + 50 μM chloroquine (CQ) to DMEM states (related to Fig 9E).** After incubation in DMEM + CQ for 60 min, medium was changed from DMEM + CQ to DMEM. Merged images (Z-stack = 2) for mKeima signals [neutral (green) + acidic (red)] were captured every 5 sec. Scale bar indicates 20 μm.
(MOV)

**S15 Movie. A time-lapse movie of MEFs expressing mKeima-LC3B under transitional conditions from nutrient-rich (DMEM) + 50 μM chloroquine (CQ) to starvation (EBSS) states (related to Fig 9F).** After incubation in DMEM + CQ for 60 min, medium was changed from DMEM + CQ to EBSS. Merged images (Z-stack = 2) for mKeima signals [neutral (green) + acidic (red)] were captured every 5 sec. Scale bar indicates 20 μm.
(MOV)

**S16 Movie. A time-lapse movie of MEFs expressing mKeima-LC3B under transitional conditions from nutrient-rich (DMEM) + 50 μM chloroquine (CQ) to starvation (EBSS) + 50 μM chloroquine (CQ) states.** After incubation in DMEM + CQ for 60 min, medium was changed from DMEM + CQ to EBSS + CQ. Merged images (Z-stack = 2) for mKeima signals [neutral (green) + acidic (red)] were captured every 5 sec. Scale bar indicates 20 μm.
(MOV)

**S1 File. Uncropped images for immunoblots.** Immunoreactivities were visualized with Immobilon-Western Chemiluminescent HRP Substrate (Millipore) and analyzed using Ez-Capture Analyzer (ATTO).
(PDF)

## Acknowledgments

We are grateful to all members of our laboratory at Tokai University School of Medicine for helpful discussion throughout this work. We thank Dr. Noboru Mizushima at The University of Tokyo for generous gift of GFP-LC3#53 transgenic mice, and all members at Support Center for Medical Research and Education at Tokai University for their technical help.

## Author Contributions

**Conceptualization:** Shinji Hadano.

**Data curation:** Masayuki Tanaka, Chisa Okada.

**Formal analysis:** Hideki Hayashi, Ting Wang, Masayuki Tanaka, Chisa Okada, Masatoshi Ito, Yumi Iida, Ayumi Sasaki, Kazuhiro Yoshida.

**Funding acquisition:** Shinji Hadano.

**Investigation:** Hideki Hayashi, Ting Wang, Masayuki Tanaka, Sanae Ogiwara, Chisa Okada, Masatoshi Ito, Nahoko Fukunishi, Yumi Iida, Ayaka Nakamura, Ayumi Sasaki, Shunji Amano, Kazuhiro Yoshida.

**Methodology:** Hideki Hayashi, Ting Wang, Masayuki Tanaka, Sanae Ogiwara, Chisa Okada, Masatoshi Ito, Nahoko Fukunishi, Yumi Iida, Ayaka Nakamura, Ayumi Sasaki, Shunji Amano, Kazuhiro Yoshida, Masato Ohtsuka.

**Project administration:** Shinji Hadano.

**Resources:** Asako Otomo, Masato Ohtsuka, Shinji Hadano.

**Supervision:** Shinji Hadano.

**Validation:** Masayuki Tanaka, Chisa Okada, Shunji Amano, Asako Otomo.

**Visualization:** Hideki Hayashi, Masayuki Tanaka, Chisa Okada, Nahoko Fukunishi, Shunji Amano, Kazuhiro Yoshida, Shinji Hadano.

**Writing – original draft:** Hideki Hayashi, Ting Wang, Masayuki Tanaka, Sanae Ogiwara, Chisa Okada, Masatoshi Ito, Nahoko Fukunishi, Yumi Iida, Ayaka Nakamura, Ayumi Sasaki, Shunji Amano, Kazuhiro Yoshida, Shinji Hadano.

**Writing – review & editing:** Asako Otomo, Masato Ohtsuka, Shinji Hadano.

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
