## [Decision Letter · Decision Letter 0]

1 Nov 2019

PONE-D-19-24727

Monitoring the autophagy-endolysosomal system using monomeric Keima-fused MAP1LC3B

PLOS ONE

Dear Dr. Hadano,

Thank you for submitting your manuscript to PLOS ONE. After careful consideration, we feel that it has merit but does not fully meet PLOS ONE’s publication criteria as it currently stands. Therefore, we invite you to submit a revised version of the manuscript that addresses the points raised during the review process.

Reviewer 1:

The authors tried to develop a new autophagy endolysosomal pathway sensor system using keima red fused to LC3.  Although mKeima-LC3B-tg mice failed to show promising results, mice derived MEF based system proved as a sensitive reporter system. They show convincing images and quantitative data to support the finding.  

1. The authors analyzed Ph sensitivity of Keima LC3 after fixation. it is better to perform this before fixation.

2. The study reveals rapid and reversible autonomous change of pH in LC3-positive autophagosomes before the vesicle-fusion. This is an interesting observation needs further validation. The methodology is having potential application in diverse areas of biology including drug screening.

Reviewer 2:

The authors have generated a mKeima-LC3B fusion construct. They have generated transgenic mouse lines expressing this construct to monitor autophagy in vivo, and also performed live imaging in vitro in mouse embryonic fibroblasts derived from the transgenic mice. They conclude that this probe may not be an appropriate tool to monitor autophagy in vivo, but could be very useful for experiments in cultured cells. 

The manuscript is well written and the figures have been made with much care, but unfortunately there are some important issues that dampen may enthusiasm:

1. It is not clear to me what the added value is of adding LC3 to the existing mKeima probe. mKeima is a well-validated autophagy probe that is taken up into autophagosomes after autophagy induction and allows distinction of autophagosomes from lysosomes. What is the advantage of attaching LC3 to mKeima? The rationale for making this fusion construct should be explained much better. 

2. It is very strange that mKeima-containing acidic vesicles (confirmed to be lysosomes by costaining with LysoTracker) become neutral within 10 seconds after switching from EBSS to DMEM. It seems very unlikely that the proton gradient across the lysosomal membrane can be completely abolished within 10 seconds after this medium switch. Which signaling pathway could possibly explain such rapid de-acidification? This finding seems to hint at some possible artifact related to the medium switch. More data would be required to make this finding more convincing.

3. The in vivo data in the transgenic mice are not adding much and could be removed from the manuscript

Minor:

What is the meaning of the mKeima-LC3’ (as opposed to mKeima-LC3) band in the western blot figures?

We would appreciate receiving your revised manuscript by Dec 16 2019 11:59PM. To enhance the reproducibility of your results, we recommend that if applicable you deposit your laboratory protocols in protocols.io, where a protocol can be assigned its own identifier (DOI) such that it can be cited independently in the future. For instructions see: http://journals.plos.org/plosone/s/submission-guidelines#loc-laboratory-protocols

We look forward to receiving your revised manuscript.

Kind regards,

Vladimir Trajkovic

Academic Editor

PLOS ONE

Journal Requirements:

Reviewers' comments:

Reviewer's Responses to Questions

**Comments to the Author**

1. Is the manuscript technically sound, and do the data support the conclusions?

Reviewer #1: Yes

Reviewer #2: No

2. Has the statistical analysis been performed appropriately and rigorously? 

Reviewer #1: Yes

Reviewer #2: Yes

3. Have the authors made all data underlying the findings in their manuscript fully available?

Reviewer #1: Yes

Reviewer #2: Yes

4. Is the manuscript presented in an intelligible fashion and written in standard English?

Reviewer #1: Yes

Reviewer #2: Yes

5. Review Comments to the Author

Reviewer #1: The authors tried to develop a new autophagy endolysosomal pathway sensor system using keima red fused to LC3. Although mKeima-LC3B-tg mice failed to show promising results , mice derived MEF based system proved as a sensitive reporter system. They show convincing images and quantitative data to support the finding.

The authors analyzed Ph sensitivity of Keima LC3 after fixation. it is better to perform this before fixation.

The study reveal rapid and reversible autonomous change of pH in LC3-positive autophagosomes before the vesicle-fusion. This is an interesting observation needs further validation. The methodology is having potential application in diverse areas of biology including drug screening.

Reviewer #2: The authors have generated a mKeima-LC3B fusion construct. They have generated transgenic mouse lines expressing this construct to monitor autophagy in vivo, and also performed live imaging in vitro in mouse embryonic fibroblasts derived from the transgenic mice. They conclude that this probe may not be an appropriate tool to monitor autophagy in vivo, but could be very useful for experiments in cultured cells.

The manuscript is well written and the figures have been made with much care, but unfortunately there are some important issues that dampen may enthusiasm:

1. It is not clear to me what the added value is of adding LC3 to the existing mKeima probe. mKeima is a well-validated autophagy probe that is taken up into autophagosomes after autophagy induction and allows distinction of autophagosomes from lysosomes. What is the advantage of attaching LC3 to mKeima? The rationale for making this fusion construct should be explained much better.

2. It is very strange that mKeima-containing acidic vesicles (confirmed to be lysosomes by costaining with LysoTracker) become neutral within 10 seconds after switching from EBSS to DMEM. It seems very unlikely that the proton gradient across the lysosomal membrane can be completely abolished within 10 seconds after this medium switch. Which signaling pathway could possibly explain such rapid de-acidification? This finding seems to hint at some possible artifact related to the medium switch. More data would be required to make this finding more convincing.

3. The in vivo data in the transgenic mice are not adding much and could be removed from the manuscript

Minor:

What is the meaning of the mKeima-LC3’ (as opposed to mKeima-LC3) band in the western blot figures?

6. PLOS authors have the option to publish the peer review history of their article (what does this mean?). If published, this will include your full peer review and any attached files.

Reviewer #1: No

Reviewer #2: No

---

## [Author Response · Author response to Decision Letter 0]

2 Jan 2020

To answer the comments and/or questions provided by the reviewers, we have conducted an additional set of experiments, whose results are exclusively included in our revised manuscript. Comments and questionnaires from reviewers are itemized and accompanied with our answers/responses as bellows.

Reviewer #1’s comments:

1. The authors analyzed pH sensitivity of Keima LC3 after fixation. it is better to perform this before fixation.

- Response to this comment:

We appreciate the reviewer’s important comment. As suggested by the reviewer, monitoring the pH sensitivity without fixation must be an ideal way to do. However, without fixation, which means the cells still alive, the incubation with acid or alkaline buffer solutions, in particular under extreme pH conditions, is harmful to cells; sometimes resulting in cell death. Under such conditions, analysis of pH sensitivity of expressed mKeima-LC3B may be extremely difficult. In addition, even after fixation, mKeima seems to keep the variable excitation spectra in a pH-dependent fashion. Thus, we are currently believing that the fixation of cells is the best way to stably measure and estimate the pH sensitivity of mKeima-LC3B molecules in situ. Alternatively, it may also be possible to biochemically measure the sensitivities using purified proteins. Indeed, the pH sensitivity of mKeima using purified protein has already been reported in the original publication (Katayama et al, Chem Biol 18: 1042, 2011). However, in our hand, purification of the mKeima-LC3B fusion protein has been unsuccessful thus far. These are the reasons why we have currently adopted the fixed cells in this study.

2. The study reveals rapid and reversible autonomous change of pH in LC3-positive autophagosomes before the vesicle-fusion. This is an interesting observation needs further validation. The methodology is having potential application in diverse areas of biology including drug screening.

- Response to this comment:

We appreciate the reviewer’s invaluable comments. As pointed out by the reviewer, we have not shown the direct evidence whether a rapid and reversible change of pH in mKeim-LC3B-positive autophagosomes occurs autonomously without committing the vesicle-fusion. Since we were unable to clearly distinguish between nascent/immature and maturated mKeima-LC3B-positive autophagosomes by staining with LysoTracker (see Fig 7), we tried to do an additional set of experiments of a live-cell imaging. Although the pharmacological inhibition of vesicular proton pumps by chloroquine (CQ) resulted in suppression of nutritional-condition-dependent rapid changes in vesicular pH (see S21 movie); yet such suppression was still reversible (see Fig 9F), we had been unable to show the commitment of vesicle fusion by this procedure. Thus, due to lack of sufficient evidences, we decided to decline the over-stated sentences that were describing “the commitment of vesicle fusion” in the phenomena of a rapid and reversible pH changes, in order to avoid the confusion as well as misunderstanding by the readers.

Reviewer #2’s comments:

1. It is not clear to me what the added value is of adding LC3 to the existing mKeima probe. mKeima is a well-validated autophagy probe that is taken up into autophagosomes after autophagy induction and allows distinction of autophagosomes from lysosomes. What is the advantage of attaching LC3 to mKeima? The rationale for making this fusion construct should be explained much better. 

- Response to this comment:

We appreciate the reviewer’s comments. Generally, macroautophagy (autophagy hereafter) is implicated in two different types of degradation; non-selective and selective degradation. Recent evidences have uncovered that dysregulation of the selective cargo-degradation system rather than non-selective one is associated with many pathological conditions such as cancer, inflammation and neurodegenerative diseases, and that LC3 and other autophagy receptor molecules containing the LIR (LC3-interacting region), such as SQSTM1/p62, NBR1 and optineurin, contributes to such disorders. Since LC3 is known to not only directly bind to autophagosomal membranes, but also simultaneously to several autophagy receptors, attaching LC3 molecule to mKeima has a strong advantage to monitor the selective-degradation activity, as has already been shown by many previous publications using GFP-LC3 fusion molecules. By contrast, in the original paper by Katayama et al (Chem Biol 2011), lone mKeima can become a probe for non-selective autophagy but not for selective ones. In this revised manuscript, we added the phrase “the selective cargo-degradation system” in the section of Introduction, to facilitate the proper understanding by the readers.

2. It is very strange that mKeima-containing acidic vesicles (confirmed to be lysosomes by containing with LysoTracker) become neutral within 10 seconds after switching from EBSS to DMEM. It seems very unlikely that the proton gradient across the lysosomal membrane can be completely abolished within 10 seconds after this medium switch. Which signaling pathway could possibly explain such rapid de-acidification? This finding seems to hint at some possible artifact related to the medium switch. More data would be required to make this finding more convincing.

- Response to this comment:

We appreciate the reviewer’s invaluable comments. First, as already mentioned in response to the Reviewer#1, we were unable to clearly distinguish between nascent/immature and maturated mKeima-LC3B-positive autophagosomes by staining with LysoTracker (Fig 7). It was also noted that LysoTracker signals were not changed by the DMEM-EBSS medium switch (revised S9 Movie). We are currently speculating that LysoTracker-labeled vesicles represent not only fully matured autolysosomes/lysosomes but also immature autophagosomes/late endosomes as well as amphisomes, which are emerged by a gradual acidification of autophagosomes/endosomes through the activation of vesicular proton pumps.

To further response to the reviewer’s questions, we have conducted an additional set of experiments. In order to test whether the medium change itself affected fluorescent signals, we performed a live-cell imaging under repeated changes of the same medium; i.e., DMEM to DMEM, or EBSS to EBSS. Although some signal fluctuations during the medium exchange were observed; which might be due to the deflection of laser-beam by transiently lowering level of medium solution, there were no observable differences in the acid/neutral signal ratio between before and after changing the medium (revised S4 Movie and S5 Movie). Therefore, medium change process itself may not affect the acidic/neutral signal ratio of mKeima-LC3B-positive compartments. Next, to test whether a rapid and reversible change in intraluminal pH of mKeima-LC3B-positive vesicles was associated with genuine maturation of autophagosomes and/or endosomes, we conducted a live-cell imaging as well as quantification of acidic/neutral signals for mKeima-LC3B in the presence or absence of CQ. All the results obtained are exclusively described in a separate paragraph of the Results section and related additional figures and movies were included (Fig 9, S15-S21 Movies). Although the deacidification by CQ proceeded much slower than did by medium switch from EBSS to DMEM, the pharmacological inhibition of vesicular proton pumps by CQ results in suppression of nutritional-condition-dependent rapid changes in vesicular pH (revised S21 movie). Importantly, such suppression was still reversible (revised Fig 9F). The results indicate that a rapidly changeable intraluminal pH of mKeima-LC3B-positive vesicles depends, at least in part, on the process of the authentic maturation system of autophagosomes and/or endosomes.

Based on our results, it is possible that continuous and persistent activation of proton pumps may be required to keep autophagosomes/endosomes towards acidic in their reversible phase. Under nutrition-rich conditions, the proton pump activity may rapidly decline via a certain signaling pathway. Consequently, the ability of maintaining an intraluminal acidity is rapidly lost, resulting in autophagosomes with neutral pH within 10 sec. Conversely, when starved, massive activation of proton pumps may explain a rapid acidification of autophagosomes. Thus far, we could not fully explain the mechanism and/or signaling pathway for these rapid-reversible phenomena in an evidence-based manner. Further studies are warranted as stated in the manuscript.

3. The in vivo data in the transgenic mice are not adding much and could be removed from the manuscript

- Response to this comment:

We appreciate the reviewer’s important suggestions. As pointed out by the reviewer, the results were rather negative indeed. However, if allowed, we would like to show those data in order to notify the negative side of mKeima molecule in the use of in vivo experiments to the readers.

4. What is the meaning of the mKeima-LC3’ (as opposed to mKeima-LC3) band in the western blot figures?

- Response to this comment:

We appreciate the reviewer’s comments. To avoid the confusion, we clearly added the definition of “mKeima-LC3’” in the legends of Fig 1, Fig 2, Fig 4 and Fig 6.

---

## [Decision Letter · Decision Letter 1]

10 Feb 2020

PONE-D-19-24727R1

Monitoring the autophagy-endolysosomal system using monomeric Keima-fused MAP1LC3B

PLOS ONE

Dear Dr. Hadano,

Thank you for submitting your manuscript to PLOS ONE. After careful consideration, we feel that it has merit but does not fully meet PLOS ONE’s publication criteria as it currently stands. Therefore, we invite you to submit a revised version of the manuscript that addresses the points raised during the review process.

Reviewer 2:

The revised manuscript has not really improved compared with the previous version. The response to my previous comments is not satisfactory.

1. It remains unclear what the added value is of adding LC3 to the existing mKeima probe. The response of the authors to this comment does not make sense. LC3 is involved in selective as well as non-selective autophagy. The response by the authors that “attaching LC3 molecule to mKeima has a strong advantage to monitor the selective degradation activity” is simply not correct.

2. The finding that mKeima-containing acidic vesicles become neutral within 10 seconds after switching from EBSS to DMEM remains very strange and the mechanism remains completely unexplained.

We would appreciate receiving your revised manuscript by Mar 26 2020 11:59PM. To enhance the reproducibility of your results, we recommend that if applicable you deposit your laboratory protocols in protocols.io, where a protocol can be assigned its own identifier (DOI) such that it can be cited independently in the future. For instructions see: http://journals.plos.org/plosone/s/submission-guidelines#loc-laboratory-protocols

We look forward to receiving your revised manuscript.

Kind regards,

Vladimir Trajkovic

Academic Editor

PLOS ONE

Reviewers' comments:

Reviewer's Responses to Questions

**Comments to the Author**

1. If the authors have adequately addressed your comments raised in a previous round of review and you feel that this manuscript is now acceptable for publication, you may indicate that here to bypass the “Comments to the Author” section, enter your conflict of interest statement in the “Confidential to Editor” section, and submit your "Accept" recommendation.

Reviewer #1: All comments have been addressed

Reviewer #2: (No Response)

2. Is the manuscript technically sound, and do the data support the conclusions?

Reviewer #1: Yes

Reviewer #2: No

3. Has the statistical analysis been performed appropriately and rigorously? 

Reviewer #1: Yes

Reviewer #2: Yes

4. Have the authors made all data underlying the findings in their manuscript fully available?

Reviewer #1: Yes

Reviewer #2: Yes

5. Is the manuscript presented in an intelligible fashion and written in standard English?

Reviewer #1: Yes

Reviewer #2: Yes

6. Review Comments to the Author

Reviewer #1: The authors have explained the details of the changes made as per the concerns raised by the reviewers. The modified version looks good.

Reviewer #2: The revised manuscript has not really improved compared with the previous version. The response to my previous comments is not satisfactory.

1. It remains unclear what the added value is of adding LC3 to the existing mKeima probe. The response of the authors to this comment does not make sense. LC3 is involved in selective as well as non-selective autophagy. The response by the authors that “attaching LC3 molecule to mKeima has a strong advantage to monitor the selective degradation activity” is simply not correct.

2. The finding that mKeima-containing acidic vesicles become neutral within 10 seconds after switching from EBSS to DMEM remains very strange and the mechanism remains completely unexplained.

7. PLOS authors have the option to publish the peer review history of their article (what does this mean?). If published, this will include your full peer review and any attached files.

Reviewer #1: No

Reviewer #2: No

---

## [Author Response · Author response to Decision Letter 1]

8 Mar 2020

March 9, 2020

Dr. Vladimir Trajkovic

Academic Editor

PLOS ONE

Dear Dr. Trajkovic,

We greatly appreciate the excellent reviews. A revision of our manuscript (PONE-D-19-24727R1) by Hideki Hayashi et al. has been completed. Please find uploaded files for our revised manuscript entitled “Monitoring the autophagy-endolysosomal system using monomeric Keima-fused MAP1LC3B” by Hideki Hayashi, Ting Wang, Masayuki Tanaka, Sanae Ogiwara, Chisa Okada, Masatoshi Ito, Nahoko Fukunishi, Yumi Iida, Ayaka Nakamura, Ayumi Sasaki, Shunji Amano, Kazuhiro Yoshida, Asako Otomo, Masato Ohtsuka and Shinji Hadano. We also attached the manuscript, in which corrections that we made were highlighted as red-colored letters, as a separate file.

To answer the comments and/or questions provided by the reviewer#2, we have tried to add more detailed explanations with additional citations mainly to the sections of “Introduction” and “Discussion”, and at the same time, to refine them as precisely as possible. Comments and questionnaires from reviewers are itemized and accompanied with our answers/responses as bellows.

Reviewer #2’s comments:

1. It remains unclear what the added value is of adding LC3 to the existing mKeima probe. The response of the authors to this comment does not make sense. LC3 is involved in selective as well as non-selective autophagy. The response by the authors that “attaching LC3 molecule to mKeima has a strong advantage to monitor the selective degradation activity” is simply not correct.

- Response to this comment:

We appreciate the reviewer’s comment. We would like to admit our misunderstanding and totally agree with what the reviewer has pointed-out; i.e., LC3 is involved both in selective and non-selective autophagy. To facilitate the readers’ proper understanding of our aim as well as reason as to why the LC3 molecule attached to mKeima, we tried to correct and explain them by adding several sentences to the section of “Introduction” (see lines 68-77 in the correction-highlighted manuscript).

2. The finding that mKeima-containing acidic vesicles become neutral within 10 seconds after switching from EBSS to DMEM remains very strange and the mechanism remains completely unexplained.

- Response to this comment:

We appreciate the reviewer’s invaluable comments. As the reviewer has pointed out, a phenomenon of the rapid deacidification (~ within 10 sec) of mKeima-LC3B-resided compartments, which is induced by changing the media from EBSS to DMEM, is also totally unexpected to us. Thus far, to the best of our knowledge, there have been no studies demonstrating the rapid deacidification of autophagosomes and/or endocytic vesicles, let alone the mechanisms of such phenomena. Although data showing such rapid deacidification are reproducibly obtained, at least, by our hand, we honestly say that we do not have enough abilities to clarify their molecular mechanisms as well as physiological significances in an experimental evidence-based manner within a designated period of this revision time. Therefore, in this revision, we decided to add more detailed explanations in the section of “Discussion”, and proposed the hypothetical mechanisms that were drawn from the current findings on the acidification and deacidification of endocytic vesicles in conjunction with the regulation of the V-ATPase functions under amino acid deplete as well as replete conditions (see lines 619-670 in the correction-highlighted manuscript). Further, we added a separated paragraph to the section of “Discussion” to explain in more detail on arising issues on LC3-labeled compartments (see lines 671-680 in the correction-highlighted manuscript).

By submitting the manuscript to PLoS ONE, we understand that “the work described has not been submitted for publication, in whole or in part, elsewhere and all the authors listed have approved the manuscript that is enclosed”. We understand that “should the submitted material be accepted for publication in PLoS ONE, we will automatically transfer the copyright to the publisher”. Further, the authors have declared that no competing interests exist.

We hope that all the changes we made meet your acceptance of the manuscript for publication in PLoS ONE. Thank you for your generous consideration. We are looking forward to hearing from you soon.

Sincerely yours,

Shinji Hadano, Ph.D.

Professor of Department of Molecular Life Sciences

Tokai University School of Medicine

143 Shimokasuya, Isehara, Kanagawa 259-1193, JAPAN

TEL: +81-463-93-1121 (ext. 2567)

FAX: +81-463-93-3965

E-mail: shinji@is.icc.u-tokai.ac.jp

---

## [Decision Letter · Decision Letter 2]

11 May 2020

PONE-D-19-24727R2

Monitoring the autophagy-endolysosomal system using monomeric Keima-fused MAP1LC3B

PLOS ONE

Dear Dr. Hadano,

Thank you for submitting your manuscript to PLOS ONE. After careful consideration, we feel that it has merit but does not fully meet PLOS ONE’s publication criteria as it currently stands. Therefore, we invite you to submit a revised version of the manuscript that addresses the points raised during the review process.

Reviewer 3:

1. A small issue with Figure 5B, the FACS plot. Four quadrants were assigned (Q1-Q4). They should explain both in text and figure legend how they are drawn and what they actually mean. 

2. Another issue is Figure 4C and 4H. Starvation induces LC3 lipidation and also p62 degradation. It's surprising to see p62 level is increased in starved samples.

We would appreciate receiving your revised manuscript by Jun 25 2020 11:59PM. To enhance the reproducibility of your results, we recommend that if applicable you deposit your laboratory protocols in protocols.io, where a protocol can be assigned its own identifier (DOI) such that it can be cited independently in the future. For instructions see: http://journals.plos.org/plosone/s/submission-guidelines#loc-laboratory-protocols

We look forward to receiving your revised manuscript.

Kind regards,

Vladimir Trajkovic

Academic Editor

PLOS ONE

Reviewers' comments:

Reviewer's Responses to Questions

**Comments to the Author**

1. If the authors have adequately addressed your comments raised in a previous round of review and you feel that this manuscript is now acceptable for publication, you may indicate that here to bypass the “Comments to the Author” section, enter your conflict of interest statement in the “Confidential to Editor” section, and submit your "Accept" recommendation.

Reviewer #3: (No Response)

2. Is the manuscript technically sound, and do the data support the conclusions?

Reviewer #3: Yes

3. Has the statistical analysis been performed appropriately and rigorously? 

Reviewer #3: Yes

4. Have the authors made all data underlying the findings in their manuscript fully available?

Reviewer #3: Yes

5. Is the manuscript presented in an intelligible fashion and written in standard English?

Reviewer #3: Yes

6. Review Comments to the Author

Reviewer #3: I'm a new reviewer so I can't answer the first question: if the authors have adequately addressed my comments in a previous round of review". However, I did see the authors addressed the comments raised by other reviewers. Although I'm not satisfied with the authors' explanation on why the acidic/neutral signal ratio of mKeima-LC3 changes so rapidly with media changes, I do recognize that it's pretty hard to clearly answer this question. I would have asked the authors to do the experiments in a couple of autophagy defective KO MEFs cells, such as ATG5, FIP200 KO etc. In addition, experiments with mito-mKeima or better with ER-mKeima maker (a recent Cell paper by Jacob Corn lab shows that starvation induces robust ER-phage) may shed more insights on this issue. But it's too much to ask for a PLOS ONE paper.

A small issue with Figure 5B, the FACS plot. Four quadrants were assigned (Q1-Q4). They should explain both in text and figure legend how they are drawn and what they actually mean.

Another issue is Figure 4C and 4H. Starvation induces LC3 lipidation and also p62 degradation. It's surprising to see p62 level is increased in starved samples.

I recommend the acceptance of this manuscript with the above suggested minor changes.

7. PLOS authors have the option to publish the peer review history of their article (what does this mean?). If published, this will include your full peer review and any attached files.

Reviewer #3: Yes: Chunxin Wang

---

## [Author Response · Author response to Decision Letter 2]

19 May 2020

May 20, 2020

Dr. Vladimir Trajkovic

Academic Editor

PLOS ONE

Dear Dr. Trajkovic,

We greatly appreciate the excellent reviews. A revision of our manuscript (PONE-D-19-24727R2) by Hideki Hayashi et al. has been completed. Please find uploaded files for our revised manuscript entitled “Monitoring the autophagy-endolysosomal system using monomeric Keima-fused MAP1LC3B” by Hideki Hayashi, Ting Wang, Masayuki Tanaka, Sanae Ogiwara, Chisa Okada, Masatoshi Ito, Nahoko Fukunishi, Yumi Iida, Ayaka Nakamura, Ayumi Sasaki, Shunji Amano, Kazuhiro Yoshida, Asako Otomo, Masato Ohtsuka and Shinji Hadano. We also attached the manuscript, in which corrections that we made were highlighted as red-colored letters, as a separate file.

To answer the comments and/or questions provided by the reviewer#3, we have slightly modified Fig 6 with the addition of some explanations both to the manuscript and legend. Further, since a new study that was related to the molecule (SLC9A6/NHE6) implicating pH modulation in endosomes had been published after the last submission of our revised manuscript, we additionally included such new findings in Discussion with the citation (as reference #35).

Reviewer #3’s comments:

1. A small issue with Figure 5B, the FACS plot. Four quadrants were assigned (Q1-Q4). They should explain both in text and figure legend how they are drawn and what they actually mean.

- Response to this comment:

We appreciate the reviewer’s invaluable comments. In a flow cytometric analysis, we determined the areas of quadrants (Q1-Q4) based on the distribution of background signals observed in wild-type MEFs. Accordingly, we newly added data showing the signal distribution of wild-type MEFs as Fig 6B, and explained how they were determined in its figure legend as well as in the corresponding section of Materials & Methods. Furthermore, to facilitate the proper understanding of the meaning of signal shifts induced by medium changes, the numerical data (showing the percentage of the cell numbers) were also included in the Result section.

2. Another issue is Figure 4C and 4H. Starvation induces LC3 lipidation and also p62 degradation. It's surprising to see p62 level is increased in starved samples.

- Response to this comment:

We appreciate the reviewer’s comment. In fact, the experimental data shown in Fig 4 are obtained under mutant SOD1-induced neurodegenerative conditions in vivo (as an ALS model), not under starved conditions. Although overexpression of SOD1 mutant (H46R) in mice can activate the autophagy, neurodegenerative stress overwhelms the capacity of protein degradation processes (not only autophagy but also the ubiquitin proteasome system), thereby progressively accumulating SQSTM1/p62, particularly, in the spinal cord (Hadano et al, PLoS ONE 2010; Hadano et al. Hum Mol Genet 2016; Mitsui et al Mol Brain 2018). As the reviewer pointed out, we have indeed observed a transient decrease of the SQSTM1/p62 levels in MEFs (in vitro) under starved conditions (see Fig 6A). However, in this study, we only analyzed the levels of LC3 (Figs 2&3), but not those of SQSTM1/p62, under starved conditions in vivo.

By submitting the manuscript to PLoS ONE, we understand that “the work described has not been submitted for publication, in whole or in part, elsewhere and all the authors listed have approved the manuscript that is enclosed”. We understand that “should the submitted material be accepted for publication in PLoS ONE, we will automatically transfer the copyright to the publisher”. Further, the authors have declared that no competing interests exist.

We hope that all the changes we made meet your acceptance of the manuscript for publication in PLoS ONE. Thank you for your generous consideration. We are looking forward to hearing from you soon.

Sincerely yours,

Shinji Hadano, Ph.D.

Professor of Department of Molecular Life Sciences

Tokai University School of Medicine

143 Shimokasuya, Isehara, Kanagawa 259-1193, JAPAN

TEL: +81-463-93-1121 (ext. 2567)

FAX: +81-463-93-3965

E-mail: shinji@is.icc.u-tokai.ac.jp

---

## [Editor Report · Decision Letter 3]

21 May 2020

Monitoring the autophagy-endolysosomal system using monomeric Keima-fused MAP1LC3B

PONE-D-19-24727R3

Dear Dr. Hadano,

We are pleased to inform you that your manuscript has been judged scientifically suitable for publication and will be formally accepted for publication once it complies with all outstanding technical requirements.

With kind regards,

Vladimir Trajkovic

Academic Editor

PLOS ONE
---

## [Editor Report · Acceptance letter]

26 May 2020

PONE-D-19-24727R3 

Monitoring the autophagy-endolysosomal system using monomeric Keima-fused MAP1LC3B 

Dear Dr. Hadano:

I am pleased to inform you that your manuscript has been deemed suitable for publication in PLOS ONE. Congratulations! Your manuscript is now with our production department. 

With kind regards,

on behalf of

Prof. Vladimir Trajkovic 

Academic Editor

PLOS ONE